# A Novel Improved Variational Mode Decomposition-Temporal Convolutional Network-Gated Recurrent Unit with Multi-Head Attention Mechanism for Enhanced Photovoltaic Power Forecasting

**Hua Fu [1], Junnan Zhang [1,\*] and Sen Xie [2]**

1   Faculty of Electrical and Control Engineering, Liaoning Technical University, Huludao 125105, China; fuhua@lntu.edu.cn
2   Institute of Intelligence Science and Engineering, Shenzhen Polytechnic University, Shenzhen 518055, China; xiesen@szpu.edu.cn
\*   Correspondence: 472010030@stu.lntu.edu.cn

**Abstract:** Photovoltaic (PV) power forecasting plays a crucial role in optimizing renewable energy integration into the grid, necessitating accurate predictions to mitigate the inherent variability of solar energy generation. We propose a novel forecasting model that combines improved variational mode decomposition (IVMD) with the temporal convolutional network-gated recurrent unit (TCN-GRU) architecture, enriched with a multi-head attention mechanism. By focusing on four key environmental factors influencing PV output, the proposed IVMD-TCN-GRU framework targets a significant research gap in renewable energy forecasting methodologies. Initially, leveraging the sparrow search algorithm (SSA), we optimize the parameters of VMD, including the mode component K-value and penalty factor, based on the minimum envelope entropy principle. The optimized VMD then decomposes PV power, while the TCN-GRU model harnesses TCN's proficiency in learning local temporal features and GRU's capability in rapidly modeling sequence data, while leveraging multi-head attention to better utilize the global correlation information within sequence data. Through this design, the model adeptly captures the correlations within time series data, demonstrating superior performance in prediction tasks. Subsequently, the SSA is employed to optimize GRU parameters, and the decomposed PV power mode components and environmental feature attributes are inputted into the TCN-GRU neural network. This facilitates dynamic temporal modeling of multivariate feature sequences. Finally, the predicted values of each component are summed to realize PV power forecasting. Validation using real data from a PV station corroborates that the novel model demonstrates a substantial reduction in RMSE and MAE of up to 55.1% and 54.5%, respectively, particularly evident in instances of pronounced photovoltaic power fluctuations during inclement weather conditions. The proposed method exhibits marked improvements in accuracy compared to traditional PV power prediction methods, underscoring its significance in enhancing forecasting precision and ensuring the secure scheduling and stable operation of power systems.

**Keywords:** photovoltaic power forecasting; gated recurrent units; minimum envelope entropy; VMD decomposition; TCN

## 1. Introduction

As fossil energy is restricted by resource reserves and environmental problems, it has become a consensus of global development to vigorously develop and efficiently utilize renewable energy [1,2]. Because of data released by the national energy administration, for solar power generation in China, the installed capacity is about 650 million kilowatts as of the end of February 2024—a year-on-year increase of 56.9%. In the future, the proportion of new energy installations will continue to increase, and the photovoltaic and other new

energy industries will need to develop rapidly. It not only provides strong guarantees for energy security but also injects new impetus into economic growth and the achievement of carbon peaking and carbon neutrality goals.

However, photovoltaic power is distinguished by its unpredictability and instability, posing significant disruptions to the regular functioning of extensive grid-connected solar photovoltaic systems and presenting considerable hurdles to the power grid's quality and stability [3–5]. This fluctuation or intermittency is caused by several factors, i.e., humidity, air pressure, irradiance, and temperature. When the meteorological factors change, large power fluctuations at the power supply side are produced in the power system, bringing operation risks to the active power balance and frequency regulation and affecting the economy of the power system [6–8]. Thus, to alleviate the problem, accurate prediction of PV power as the key technology is present. Meanwhile, it supplies guidance for unit commitment, thereby reducing the power generation cost and strengthening the competitiveness of photovoltaic energy in the electricity market. Hence, reliable photovoltaic forecasting technology of the power system is crucial for the economical and safe operation and photovoltaic field management.

Currently, numerous studied on photovoltaic power forecasting have been carried out, and the forecasting approaches have been presented as time series [9–11], neural networks [12–14], support vector machines [15,16], Markov chain [17,18] and a combination of corresponding methods [19–21]. As the scale of power plants continues to expand, the amount of data produced by power plants has also exploded. In fact, due to the quantity and quality of the source data of power plants, the traditional neural network photovoltaic power forecasting model is restricted by not considering environmental factors [22], thereby lacking reasonable utilization of complex sequence information. In addition, considering the nonlinear change in photovoltaic power and multiple environment sequence information, the convergence rate of the model slows down and overfitting appears with the increase in network input variables [23–25]. At the same time, the accuracy of photovoltaic power forecasting is also affected by time-varying factors [26,27]. Therefore, to guarantee the feasibility of photovoltaic power forecasting, it is beneficial to fully analyze the impact of environmental factors on the modeling of photovoltaic power forecasting. Moreover, the long short-term memory (LSTM) network, as referenced in the literature [28,29], represents a type of deep neural network. Within the framework of deep learning models, the LSTM network stands out for its exceptional proficiency in addressing issues related to time series forecasting, attributable to its distinctive architectural design. But, the LSTM architecture is characterized by a higher parameter count compared to the GRU architecture. Specifically, LSTM incorporates a greater number of gating units and parameters, resulting in increased model complexity and computational demands. This elevated parameterization in LSTM models may consequently lead to escalated training and inference costs. GRU has been identified as a well-suited solution for managing and predicting the challenges associated with extended time intervals and temporal delays within time series data. Its efficacy in addressing such issues has led to widespread adoption across various industrial processes. In addition, time series modeling has been studied and explored in terms of photovoltaic power forecasting [30]. However, GRU network performance is greatly affected by parameters, and whether the model parameters are reasonable has a great influence on the forecasting results [31,32].

Moreover, due to some random factors such as weather, there are many uncertainties in the actual photovoltaic sequence. In addition, the photovoltaic power has non-stationary and nonlinear characteristics, and a single forecasting model is insufficient to satisfy forecasting accuracy requirements. The hybrid forecasting model with a decomposition algorithm effectively reduces the original photovoltaic sequence characteristics and has better forecasting performance. There are familiar decomposition approaches which include the variational mode, empirical mode [33], and wavelet decomposition [34]. Nevertheless, the selection of base functions and thresholds is depended on the WD effect. EMD and its derived methods lack a mathematical theoretical foundation due to endpoint effects.

VMD can effectively suppress noise and is regarded as the most effective decomposition technique. Nevertheless, intrinsic mode functions (IMFs) and the number of modes of VMD have a remarkable effect on the decomposition effect.

The above methods provide inspiration and motivation for the proposed forecasting strategy in this paper. However, the actual photovoltaic power is greatly interfered by the external environment and has the characteristics of instability and obvious intermittent fluctuation [35,36]. In addition, there are differences in photovoltaic power forecasting models under different environments, and a single model cannot meet the actual production needs. Moreover, the forecasting accuracy is directly affected by whether the selection of forecasting model parameter is reasonable. Therefore, a novel hierarchical VMD-TCN-GRU multi-head attention mechanism for photovoltaic power forecasting is present in this paper. The main innovation points of this article are as follows:

(1) To decompose photovoltaic power, the variational mode decomposition method is used. Meanwhile, the optimal mode and penalty factor are searched based on the minimum envelope entropy to enhance the adaptability of the variational mode decomposition algorithm.

(2) Different TCN-GRU models are constructed for different PV modal components decomposed via the improved variational mode decomposition algorithm, and the main environmental factors, for example, atmospheric pressure, air temperature, solar irradiance, and component temperature, are considered as TCN-GRU model inputs.

(3) According to the SSA, the hidden layer neural element number, training frequency, and learning rate parameters that have a significant impact on network performance were optimized. The forecasting results under multiple photovoltaic modes are integrated to obtain better photovoltaic power forecasting. Finally, for the proposed forecasting strategy, the photovoltaic power of the actual power plant is applied to illustrate the feasibility.

The remaining parts of this article include Section 2, which is dedicated to the detailed exposition of the methodology employed in our study, elucidating the theoretical framework and computational techniques utilized in our research investigation. Section 3 comprehensively describes the simulation results derived from the application of the proposed methodology, presenting empirical data and analysis to support our research findings. In Section 4, a rigorous discussion is conducted to evaluate and interpret the significance of the obtained results, thereby validating the rationality of our research approach and its implications for the field of study. Finally, the conclusions drawn from our research endeavor are summarized in Section 5, encapsulating the key findings, implications, and potential avenues for future research exploration.

## 2. Materials and Methods

### 2.1. TCN Network

The TCN represents a convolutional neural network architecture tailored for addressing time-series problems, integrating dilated causal convolution (DCC) and residual connection (RC) mechanisms. This architecture effectively captures the interdependencies between data points, facilitating subsequent predictions.

Dilated convolution, a key component of TCN, expands the receptive field by selectively skipping portions of the input. By adjusting the dilation factor, dilated convolution modulates the size of the receptive field, enabling the network to flexibly control the historical information incorporated into the output. In the context of one-dimensional sequential data $x \in R^n$ and filters $f : \{0, 1, \cdots, k-1\} \to R$, the convolutional kernel, characterized by filter coefficients k and dilation factor d, extends the receptive field. The operation of dilated convolution is expressed as follows:

$$F(x) = \sum_{i=0}^{k-1} f(i) \cdot x_{s-d \cdot i} \tag{1}$$

where $d$ denotes the dilation factor; $s - d \cdot i$ represents historical data in the input sequence; and $k$ stands for the filter coefficient [37].

The dilated causal convolution, as illustrated in Figure 1, reveals that the receptive field size of a point $Y_t$ in the output sequence is modulated via $k$ and $d$. Importantly, the output at a given point is influenced solely by the preceding historical data. The TCN architecture employed in this study utilizes dilated causal convolutions with dilation factors $d$ set to 1, 2, 4, and 8 and a filter coefficient $k$ of 3, as depicted in Figure 2. By flexibly adjusting the receptive field, the model comprehensively considers temporal features within the power data. Tailoring the memory length of output nodes based on varying input time scales effectively addresses the issue of historical data neglect observed in traditional methods. This adaptability proves advantageous, particularly in the context of short-term photovoltaic forecasting.

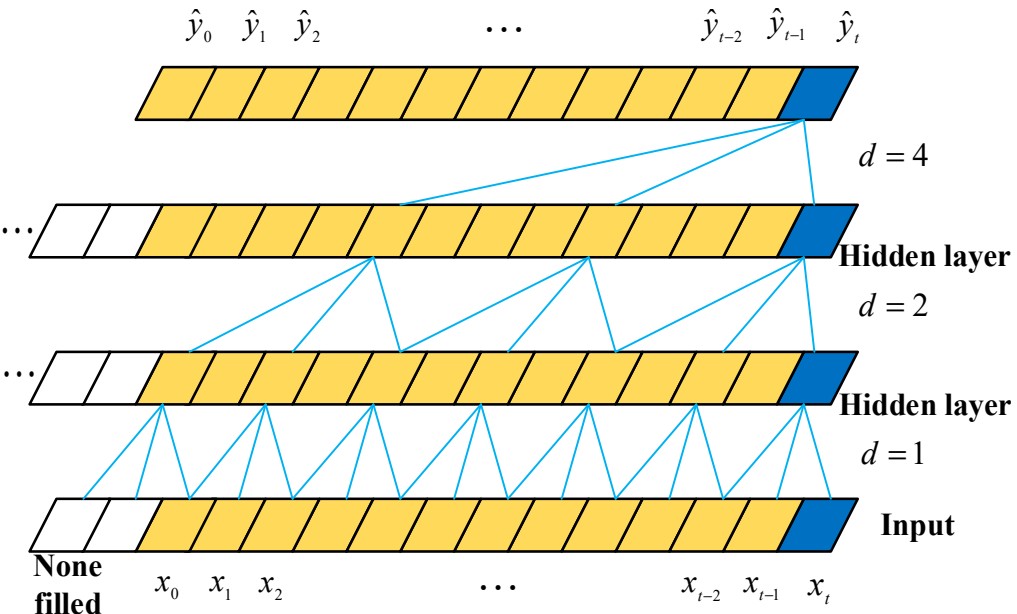

**Figure 1.** Structure diagram of dilated causal convolution network.

The residual connection has been demonstrated as an effective approach in training deep neural networks, enabling the network to propagate information across layers. The construction of a residual block to replace a single convolutional layer is depicted in the following Figure 2. A residual block comprises two convolutional layers and non-linear mapping, with WeightNorm and Dropout incorporated at each layer for network regularization.

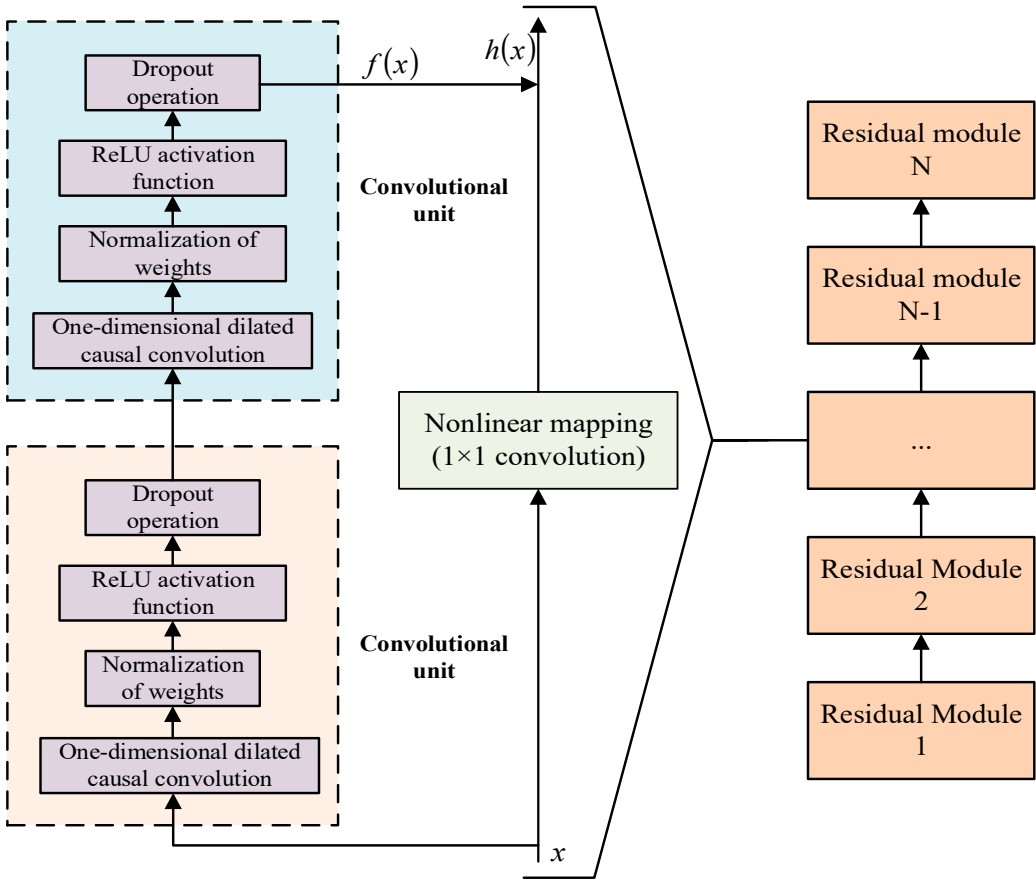

**Figure 2.** TCN residual unit structure diagram.

## 2.2. TCN-GRU

The traditional GRU network architecture represents a specialized form of recurrent neural network (RNN) models, offering an optimized alternative based on the LSTM network structure. While maintaining comparable network accuracy, the GRU achieves simplification of unit complexity by adjusting the structures of the input gate, forget gate, and output gate within the LSTM computational framework. Notably, the integration of the LSTM forget gate and input gate into a unified update gate contributes to a reduction in the model's learning and training duration, enhancing overall computational efficiency. In the GRU unit structure, the reset gate regulates the proportion of historical moment memory values entering the output gate, while the update gate determines the quantity of retained historical moment memory information, thereby governing the state update of the hidden layer. The unit structure is illustrated in Figure 3. This optimization in unit architecture exemplifies the GRU's ability to streamline computational complexity while preserving the essential memory dynamics, showcasing its potential for expedited learning and improved operational efficiency compared to conventional LSTM models.

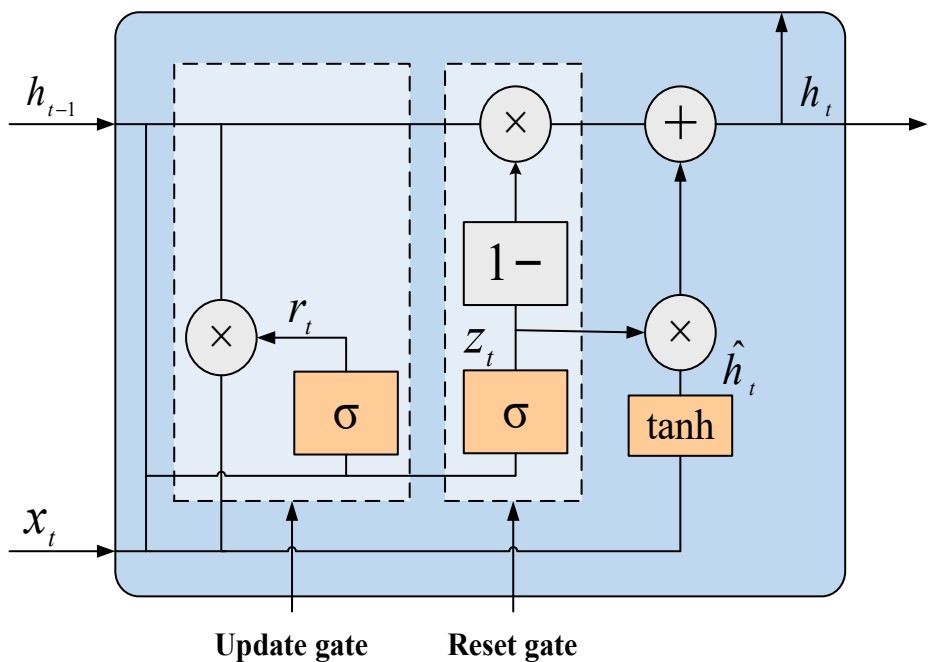

**Figure 3.** Structure diagram of GRU.

The expressions for each gate function are as follows:

$$r_t = \sigma(U_r h_{t-1} + W_r x_t + b_r), \tag{2}$$

$$z_t = \sigma(U_z h_{t-1} + W_z x_t + b_z), \tag{3}$$

$$\hat{h}_t = \tanh[W_h x_t + U_h(r_t \odot h_{t-1}) + b_h], \tag{4}$$

$$h_t = (1 - z_t) \odot h_{t-1} + z_t \odot \hat{h}_{t-1} \tag{5}$$

where $h_{t-1}$ represents the model's output at the previous time step; $x_t$ denotes the input at time $t$; $\sigma(\cdot)$ signifies the activation function, typically modeled as the sigmoid function; $r_t$ and $z_t$ denote the reset gate and update gate, respectively; $h_t$ signifies the state output of the model at time $t$; $W$ and $U$ represent weight matrices; tanh denotes the hyperbolic tangent function; $\odot$ represents the Hadamard product between two matrices; and $b$ signifies the bias terms for each input [38].

In the temporal convolutional network, the application of DCC allows TCN to possess a larger receptive field with fewer layers, enabling it to process longer historical data. The DCC utilizes an activation function and undergoes weight normalization and regularization operations. Meanwhile, the RC ensures stability in TCN by employing skip connections from the input to the output, especially in deeper TCN architectures.

The gated recurrent unit, distinct from traditional RNNs, introduces changes to the hidden layer architecture, incorporating memory cells, an update gate, and related gates. GRU determines when to update memory cells with candidate values through the update gate. Compared to one-dimensional convolutional neural networks (CNNs), TCN, due to its use of DCC and RC, can process longer historical data with increased stability. Therefore, TCN is selected for high-dimensional feature extraction from input data. For time series prediction, GRU demonstrates performance almost equivalent to LSTM but with faster training speeds, justifying its selection for sequence prediction tasks. In summary, this paper proposes a PV power prediction framework, VMD-TCN-GRU, based on VMD and incorporating TCN and GRU components. The workflow involves initial data preprocessing steps such as data cleaning and standardization, followed by the application of VMD decomposition to the processed data. Subsequently, the individual VMD modes are input into TCN residual blocks for high-dimensional feature extraction. Finally, the output from

the TCN residual blocks is fed into the GRU network for prediction, yielding the final forecast results.

*2.3. Multi Head Attention Mechanism*

While the GRU demonstrates excellent performance in sequence prediction tasks, it is not immune to the issue of error accumulation. Photovoltaic power data, being continuous over time, are subject to considerable uncertainty due to external environmental factors and unforeseen events. The impact of abrupt data changes amplifies errors over multiple time steps during training, leading to suboptimal prediction outcomes. Attention mechanisms, capable of adaptively capturing global dependencies within the data, offer the advantage of focusing not only on the information at the current position in the sequence but also on information at other positions. However, attention mechanisms necessitate the computation of weight relationships between each sequence, leading to significant computational resource requirements, especially when dealing with long sequences.

To address these challenges, this paper introduces a novel structure called a multi-head attention gated recurrent unit (MAGRU), combining the strengths of GRU and multi-head attention mechanisms. The MAGRU structure is presented in Figure 4. In this approach, a sliding window is introduced at the position of the GRU output hidden state $h_t$, aggregating the information from the preceding m time steps into a new sequence. Here, h denotes the dimensionality of the hidden state $h_t$. Subsequently, multiple sets of learnable weight matrices $W_i^Q$, $W_i^K$, and $W_i^V \in R^{h \times d}$ are introduced for each head, serving as Query, Key, and Value matrices, respectively, where *i* represents the group index, and d is the dimensionality of the attention mechanism. The calculations for the Query, Key, and Value matrices are formulated as follows [39]:

$$Q_i = HW_q, K_i = HW_k, V_i = HW_v \tag{6}$$

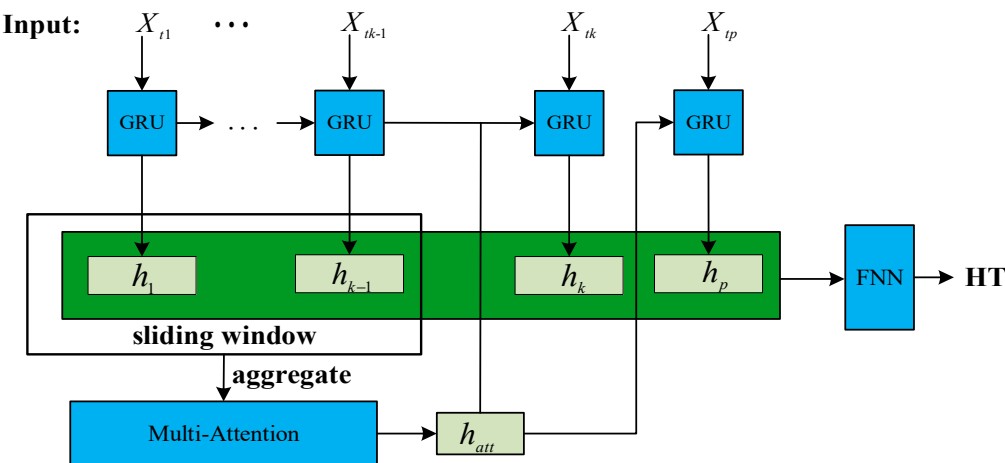

**Figure 4.** Diagram of multi-head attention GRU.

For each hidden state at time step *t*, use the Query, Key, and Value matrices to calculate its attention weight. The formula is as follows:

$$Attention_i(Q_i, K_i, V_i) = \text{softmax}(\frac{Q_i K_i^T}{\sqrt{d_a}})V_i \tag{7}$$

where $\sqrt{d_a}$ is used to scale the size of the inner product and avoid the input of the softmax function being too large or too small [40]. Concatenate the output vectors of multiple attention heads to obtain matrix *Z*:

$$Z = concat(Attention_1, \ldots, Attention_i). \tag{8}$$

Perform a linear transformation on matrix $Z$ to obtain attention score output $h_{att}$ as follows:

$$h_{att} = Z \cdot W \in R^{m \times h} \tag{9}$$

where $W$ is a learnable weight matrix. $h_{att}$ will participate in training as the hidden state of the input for the next time step GRU [41].

Finally, the obtained output is passed on to a feedforward neural network FNN to adjust its dimension and obtain the temporal feature $HT \in R^{Q \times N \times D}$ of PV power data.

### 2.4. Forecasting Modeling Based on Improved TCN-GRU

Since the actual photovoltaic power system is characterized by instability and intermittent fluctuations, to guarantee the precision and feasibility for photovoltaic power forecasting, an improved TCN-GRU network forecasting framework is proposed, as shown in Figure 5. Firstly, the photovoltaic power is decomposed using VMD, and the penalty factor and decomposition mode number of VMD are determined according to the minimum envelope entropy principle. Subsequently, daily environmental data and photovoltaic power data are utilized as TCN-GRU network inputs. In addition, the TCN-GRU is established for photovoltaic power forecasting under different modal components, and the network parameters are optimized using the SSA algorithm. Finally, to obtain final photovoltaic power forecasting, the forecasting results of the SSA-TCN-GRU model under different modal components are reconstructed.

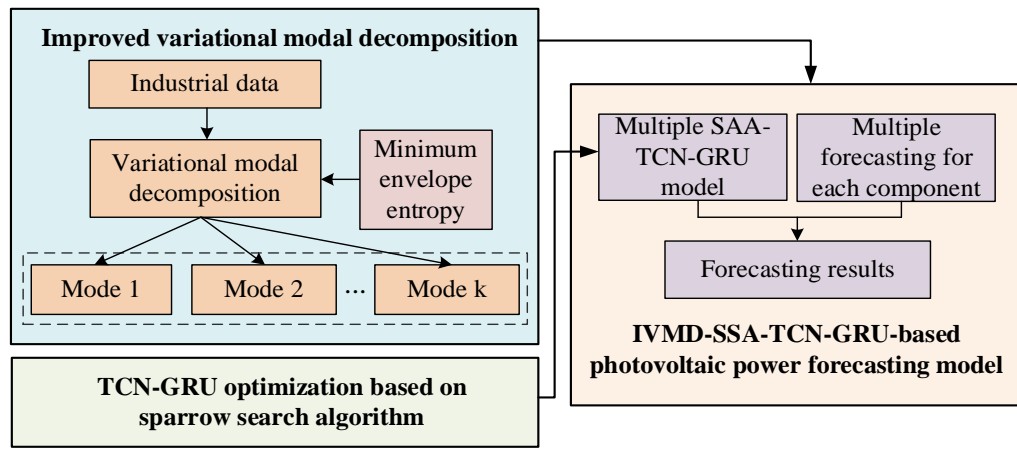

**Figure 5.** The framework of improved TCN-GRU network forecasting.

### 2.5. Improved Variational Modal Decomposition
2.5.1. Preliminary of VMD

The original photovoltaic power sequence is decomposed to obtain the separation of stationary series and non-stationary series, reduce the randomness and non-stationary of photovoltaic power, and decrease the interference in the forecasting process. The VMD represents a variational approach rooted in the amalgamation of frequency mixing, Hilbert transform, and classical Wiener filter methodologies. Distinguished by its non-recursive nature and adaptive signal processing capabilities, VMD emerges as a robust method for addressing signal decomposition challenges. Thus, original photovoltaic power signal $f(t)$ is decomposed through the VMD algorithm into $k$ discrete photovoltaic power mode $u_k(t)$, that is, the signal is decomposed into a limited number of mode components with different IMFs. Compared to empirical mode decomposition, the endpoint effects and modal aliasing problems are overcome using the VMD method.

The specific construction steps are present as follows:

1. For each mode function, through Hilbert transform, the analytic signal $u_k(t)$ is obtained to acquire its unilateral spectrum $[\delta(t) + \frac{j}{\pi t}] * u_k(t)$;

2. The frequency spectrum for each mode $u_k(t)$ is modulated to the corresponding base band $[(\delta(t) + \frac{j}{\pi t}) * u_k(t)]e^{-j\omega_k t}$ by mixing the exponential terms $e^{-j\omega_k t}$ of its corresponding center frequency $\omega_k$;

3. Through the Gaussian smoothness of the demodulation signal, for each mode signal, the bandwidth is estimated, and the constrained variational problem is obtained.

The extended Lagrange expression is as follows [42]:

$$
\begin{cases}
\min\limits_{\{u_k\},\{\omega_k\}} \left\{ \sum_{k=1}^{k} \left\| \partial_t \left[ \left( \delta(t) + \frac{1}{\pi t} \right) * u_k(t) \right] e^{-j\omega_k t} \right\|_2^2 \right\} \\
s.t. \sum\limits_{k} u_k = f
\end{cases}
. \tag{10}
$$

By introducing Lagrange multiplication operator $\lambda$ and quadratic penalty factor $\alpha$, an augmented Lagrange expression is present whereby the constrained variation is converted to the unconstrained one, as follows:

$$
\begin{aligned}
L(\{u_k\}, \{\omega_k\}, \lambda) &= \alpha \sum_k \left\| \partial_t \left[ \left( \delta(t) + \frac{j}{\pi t} \right) * u_k(t) \right] e^{-j\omega_k t} \right\|_2^2 \\
&+ \left\| f(t) - \sum_k u_u(t) \right\|_2^2 + \left\langle \lambda(t), f(t) - \sum_k u_k(t) \right\rangle
\end{aligned}
. \tag{11}
$$

The solution of each mode function is as follows:

$$
\hat{u}_k^{n+1}(\omega) = \frac{\hat{f}(\omega) - \sum\limits_{i \neq k} \hat{u}_i(\omega) + \frac{\hat{\lambda}(\omega)}{2}}{1 + 2\alpha(\omega - \omega_k)^2}. \tag{12}
$$

The solution of the center frequency for each mode is as follows:

$$
\omega_k^{n+1} = \frac{\int_0^\infty \omega |\hat{u}_k(\omega)|^2 d\omega}{\int_0^\infty |\hat{u}_k(\omega)|^2 d\omega}. \tag{13}
$$

$\lambda$ is updated as follows:

$$
\hat{\lambda}_k^{n+1}(\omega) \leftarrow \hat{\lambda}^n(\omega) + \tau \left( \hat{f}(\omega) - \sum_k \hat{u}_k^{n+1}(\omega) \right). \tag{14}
$$

Determine whether the termination condition is met.

$$
\sum_k \left\| \hat{u}_k^{n+1} - \hat{u}_k^n \right\|_2^2 \Big/ \left\| \hat{u}_k^n \right\|_2^2 < \varepsilon. \tag{15}
$$

The specific flow of the VMD algorithm is shown in Algorithm 1. Compared to EMD, VMD has a strict mathematical model and often has better robustness in dealing with noise. VMD not only effectively decomposes various harmonics but also does not consider the relative amplitude between harmonics and the distance between their respective center frequencies during mode separation. Moreover, the VMD method has high decomposition accuracy, fewer decomposition layers and no mode aliasing [43]. For photovoltaic power, it can be used as an effective means to accurately decompose the various frequency components, which is conducive to photovoltaic power forecasting under unstable and intermittent fluctuation conditions.

---

**Algorithm 1** The process of the VMD algorithm.

---

Complete Optimization of VMD
Initialize $\{\hat{u}_k^1\}$, $\{\omega_k^1\}$, $\{\hat{\lambda}_k^1\}$, $n \leftarrow 0$
Repeat
$n \leftarrow n + 1$
for k = 1: K do
Update $\hat{u}_k$ for all $\omega \geq 0$:

$$\hat{u}_k^{n+1}(\omega) \leftarrow \frac{\hat{f}(\omega) - \sum_{i<k} \hat{u}_i^{n+1}(\omega) - \sum_{i>k} \hat{u}_i^n(\omega) + \frac{\hat{\lambda}^n(\omega)}{2}}{1 + 2\alpha(\omega - \omega_k^n)^2}$$

Update $\omega_k$:

$$\omega_k^{n+1} = \frac{\int_0^\infty \omega |\hat{u}_k^{n+1}(\omega)|^2 d\omega}{\int_0^\infty |\hat{u}_k^{n+1}(\omega)|^2 d\omega}$$

end for
Dual ascent for all $\omega \geq 0$:

$$\hat{\lambda}_k^{n+1}(\omega) \leftarrow \hat{\lambda}^n(\omega) + \tau\left(\hat{f}(\omega) - \sum_k \hat{u}_k^{n+1}(\omega)\right)$$

Until convergence: $\sum_k \|\hat{u}_k^{n+1} - \hat{u}_k^n\|_2^2 \big/ \|\hat{u}_k^n\|_2^2 < \varepsilon$

---

2.5.2. VMD with Minimum Envelope Entropy

In fact, decomposition mode number k has a certain degree of impact on the decomposition effect of VMD. Moreover, in multiple IMFs, mode overlap occurs when the same component appears if the *k* is big. On the contrary, in the same IMF, there are multiple components. So far, the empirical value has been used by most mode classification studies. To solve this problem, envelope entropy is introduced to search for the optimal mode number *k* and equilibrium factor, and an improved VMD is proposed. The detailed steps for IVMD are presented:

Step 1: Input the signal $x(i)$ $(i = 1, 2, \ldots, N)$ into IVMD model.

Step 2: Initialize parameters; let $k = 2$. In addition, IVMD is performed.

Step 3: Calculate the envelope entropy, which is represented as Equation (16), and determine whether it meets the condition of minimum envelope entropy under the condition of meeting the error limit.

$$\begin{cases} E_p = -\sum_{i=1}^N p_i \log p_i \\ p_i = a(i) \big/ \sum_{i=1}^N a(i) \end{cases} \tag{16}$$

where the entropy value of $p_i$ is the envelope entropy $E_p$, and $N$ is the number of sampling points. Decomposed using VMD after Hilbert demodulation, $a(i)$ is the envelope signal of $k$ mode components; $p_i$ is the probability distribution sequence [44].

Step 4: Stop decomposition and obtain *k* when the condition is reached. Otherwise, $k = k + 1$, until the condition is satisfied.

*2.6. GRU Optimization Using SSA*

The forecasting performance according to the GRU network is greatly affected by its parameters. To fully demonstrate the effectiveness of GRU, a GRU model is established for each mode component, respectively. In addition, the number of hidden layer neurons is optimized through the SSA, training times, and the learning rate for GRU network model parameters. The SSA originates from the anti-capture and foraging behavior.

(1)    The position $X_{i,j}^{t+1}$ of the founder is updated as follows:

$$X_{i,j}^{t+1} = \begin{cases} X_{i,j}^t \exp(-i/\alpha \cdot iter_{\max}) & , R_2 < ST \\ X_{i,j}^t + Q \cdot L & , R_2 \geq ST \end{cases} \tag{17}$$

where $iter_{\max}$ represent the maximum number of iterations and $t$ denotes the current iteration number. The position of the $i$th sparrow in dimension $j$ is represented by $X_{i,j}^t$; the range of $j$ is $\{1, 2, \cdots, d\}$. Additionally, $ST \in [0.5, 1]$ and $R_2 \in [0, 1]$ refer to the security threshold and alarm value, respectively. A random number $\alpha \in (0, 1]$ is utilized in the algorithm, and $L(1 \times d)$ is a complete matrix with all elements equal to 1. A normally distributed random number, denoted as $Q$, signifies the absence of predators, prompting the finder to engage in wide-area search mode when the condition $R_2 < ST$ is met. Conversely, if there is impending danger $R_2 > ST$, all sparrows swiftly relocate to alternative safe locations [45].

(2) The position of the joiner sparrow is expressed as follows:

$$X_{i,j}^{t+1} = \begin{cases} Q \cdot \exp\left((X_{\text{worst}} - X_{i,j}^t)/i^2\right), i > n/2 \\ X_p^{t+1} + \left|X_{i,j}^t - X_p^{t+1}\right| \cdot A^+ \cdot L, \text{others} \end{cases} \tag{18}$$

where $X_{worst}$ denotes the global worst position, $X_p$ represents the optimal position of the current discoverer, and a 1xd matrix $A$ is defined with elements randomly assigned 1 or $-1$: $A^+ = A^{\text{T}}\left(AA^{\text{T}}\right)^{-1}$. The condition where $i > n/2$ indicates that the participant with poor fitness is at a heightened risk of starvation [46].

(3) Assuming that 10%–20% of the sparrow population perceives danger and promptly relocates to a safe area, the guard position $X_{i,j}^{t+1}$ is determined as follows:

$$X_{i,j}^{t+1} = \begin{cases} X_{\text{best}}^t + \beta \cdot \left|X_{i,j}^t - X_{\text{best}}^t\right|, \ f_i > f_g \\ X_{i,j}^t + K \cdot \left(\frac{\left|X_{i,j}^t - X_{\text{worst}}^t\right|}{(f_i - f_w) + \varepsilon}\right), \ f_i = f_g \end{cases} \tag{19}$$

where $\beta$ follows a normal distribution with mean 0 and variance 1, representing a random number for the step control parameter. Other variables include $X_{best}$ as the global optimal position; $K \in [-1, 1]$ as the step control parameter, denoting the direction of sparrow movement; $f_g$ as the global optimal fitness value; $f_i$ as the fitness value of individual sparrows at the current step; and $f_w$ as the global worst fitness value. To prevent division by zero, $\varepsilon$ is introduced as a minimum constant. In addition, sparrows located at the edges of the population face increased vulnerability to predators when $f_i > f_g$. Conversely, sparrows positioned in the middle of the population effectively communicate the awareness of danger to other sparrows, thereby reducing the risk of predation under the condition $f_i = f_g$ [47].

In the context of optimizing the parameters of a GRU using the SSA, a systematic procedure is outlined as follows:

Step 1: Initialization of parameters

The initial steps involve setting up the number of iterations, determining the ratio of predators within the population, and initializing the population size.

Step 2: Fitness evaluation and sorting

Subsequently, the fitness value of each individual sparrow is computed, and the population is sorted in descending order based on their fitness values.

Step 3: Update of discoverer's location

The position of the discoverer, representing the sparrow with the optimal fitness value, is then updated according to the SSA algorithm.

Step 4: Update of joiner's location

Similarly, the location of the joiner, denoting a sparrow seeking to improve its fitness by joining the discoverer, is adjusted based on the algorithm's principles.

Step 5: Update of vigilant position

The vigilant position, indicating a sparrow aware of potential danger, is updated in accordance with the algorithm's specifications.

Step 6: Fitness calculation and position update

The fitness value of the sparrows is recalculated, and their positions are updated iteratively to enhance the optimization process.

Step 7: If the stop conditions is met, the command output is displayed. Otherwise, repeat Step 2 to Step 6.

### 2.7. IVMD-SSA-TCN-GRU-Based Photovoltaic Power Forecasting Strategy

Since the intermittent fluctuation and instability characteristics for photovoltaic power, the photovoltaic power time series are firstly decomposed into different modes, and then a TCN-GRU model optimized using the SSA is established for each mode. Furthermore, the forecasting results for each mode are integrated to achieve power forecasting. Figure 6 presents the flowchart of the IVMD-SSA-TCN-GRU photovoltaic power forecast, and the specific forecasting steps are demonstrated as follows.

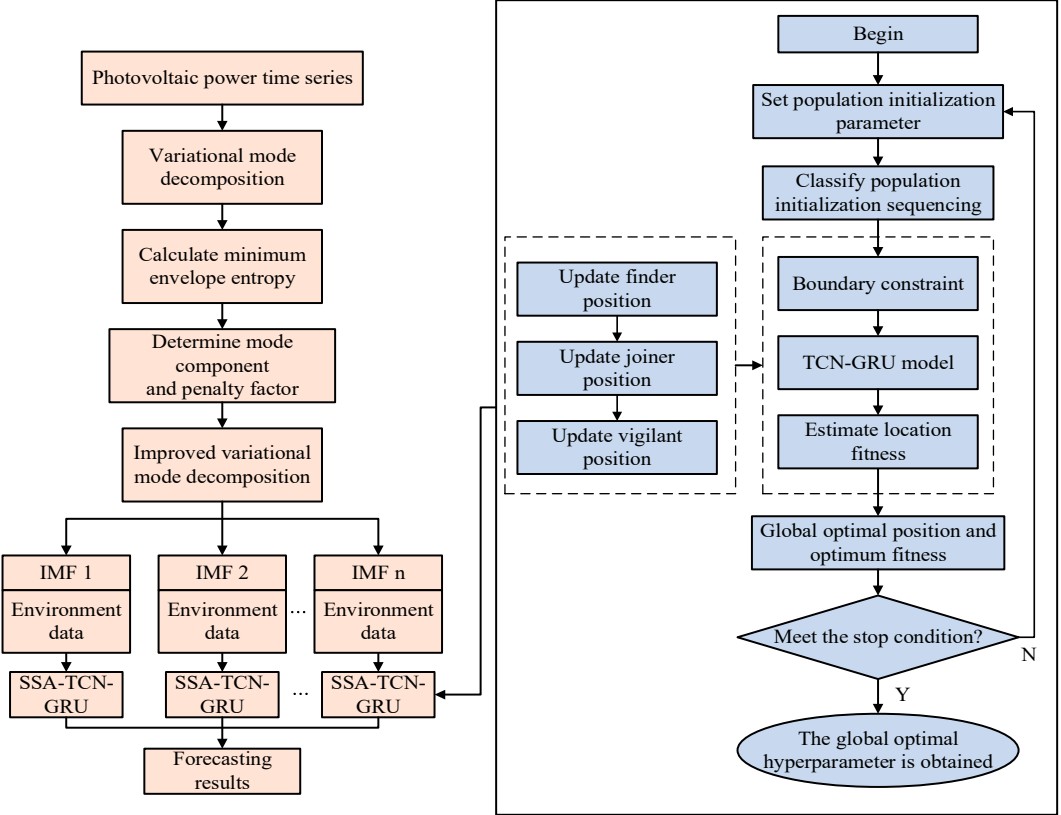

**Figure 6.** The flowchart of IVMD-SSA-TCN-GRU photovoltaic power forecasting.

Step 1: Select the environment information as the model input.

Step 2: Use the IVMD method to decompose the original photovoltaic power sequence and obtain the $k$ components.

Step 3: Firstly, set the parameter range (number of learning rate $\eta$, training times $E$, and hidden layer neurons $H$), search range of sparrow population size $N$, and maximum number of iterations $M$. Moreover, set the mean square error as an objective function for the optimization algorithm. Furthermore, set up the coupling model of SSA-TCN-GRU.

Step 4: Establish SSA-TCN-GRU forecasting models for each component and obtain $k$ forecasting models.

Step 5: Add the corresponding forecasted values of $k$ forecasting models to obtain the forecasting result of photovoltaic power.

## 3. Results

To illustrate the feasibility of the photovoltaic power forecasting strategy proposed for a photovoltaic power station in China, the historical photovoltaic power data are used for case study analysis. Photovoltaic datasets with different weather conditions (sunny, cloudy, and rainy) were selected with 864 samples, of which the first 605 were used for training and the last 259 were used for forecasting. The sampling interval was 5min, and the installed capacity of the photovoltaic field was 603 MW.

### 3.1. Data Processing

Since PV power and meteorological data contain outliers and missing values, the data error interference affects the accuracy of the models. Therefore, data reconciliation was used to clean the data, eliminate outliers, and fill in missing values [48,49].

Due to the difference in dimensionality of data for different variables, the data were standardized to ensure normal calculation.

$$\widetilde{x}(i) = \frac{x(i) - x_{\min}}{x_{\max} - x_{\min}} \tag{20}$$

where $x(i)$ is the sample of the original photovoltaic sequence or meteorological sequence; $\widetilde{x}(i)$ represents the normalized processed sequences, which are in [0, 1]; and for the sample data, $x_{\max}$ and $x_{\min}$ are the maximum and minimum value of the sample data, respectively.

### 3.2. Evaluation Index of Forecasting Model

As forecasting evaluation indexes, MAE and RMSE are utilized to quantitatively analyze the forecasting performance and generalization ability of the model.

$$\text{RMSE} = \sqrt{\frac{1}{N}\sum_{i=1}^{N}(y_i - \hat{y}_i)^2} \tag{21}$$

$$\text{MAE} = \frac{1}{N}\sum_{i=1}^{N}|y_i - \hat{y}_i| \tag{22}$$

where $N$ is the testing sample; $y_i$ is the true data of photovoltaic power; $\hat{y}_i$ is the forecasting result of photovoltaic power; and for the actual photovoltaic power, $\overline{y}$ is the average value.

### 3.3. Simulation Analysis

In addition, air temperature, irradiance, air pressure, and module temperature as well as historical photovoltaic power are determined as the IVMD-TCN-GRU forecasting model inputs. For predetermined parameters $K$ and $\alpha$, to solve the problem in the traditional VMD algorithm, an adaptive IVMD algorithm is proposed based on envelope entropy. Moreover, the non-stationary and nonlinear characteristics are decomposed for photovoltaic power. Figure 7 illustrates the iterative process of envelope entropy values under different weather conditions, and the optimal parameter combination and corresponding minimum envelope entropy obtained from this are shown in Table 1. The decomposition results of photovoltaic power corresponding to the optimal $K$ value under various weather conditions are shown in Figure 8. The IMF indicates that sub modes are obtained after photovoltaic power decomposition. The different modes after VMD decomposition not only have stronger stationarity, but also maintain the trend characteristics of original photovoltaic data well. Considering the chaotic nature of photovoltaic weather processes and the nonlinear relationship between photovoltaic weather and output photovoltaic power, TCN-GRU photovoltaic power forecasting models are established for each mode component decomposed using IVMD. In addition, to acquire the forecasted photovoltaic power, the forecasted output of each model is added. The five hyperparameters of the model are determined, that is, the number of hidden layers is 1, the dimension of the input layer is

24, the time step of the input layer is 10, the dimension of the output variable is 1, and the dimension of each hidden layer is 10. Finally, Figure 9 presents the experimental results of photovoltaic power prediction based on IVMD-TCN-GRU.

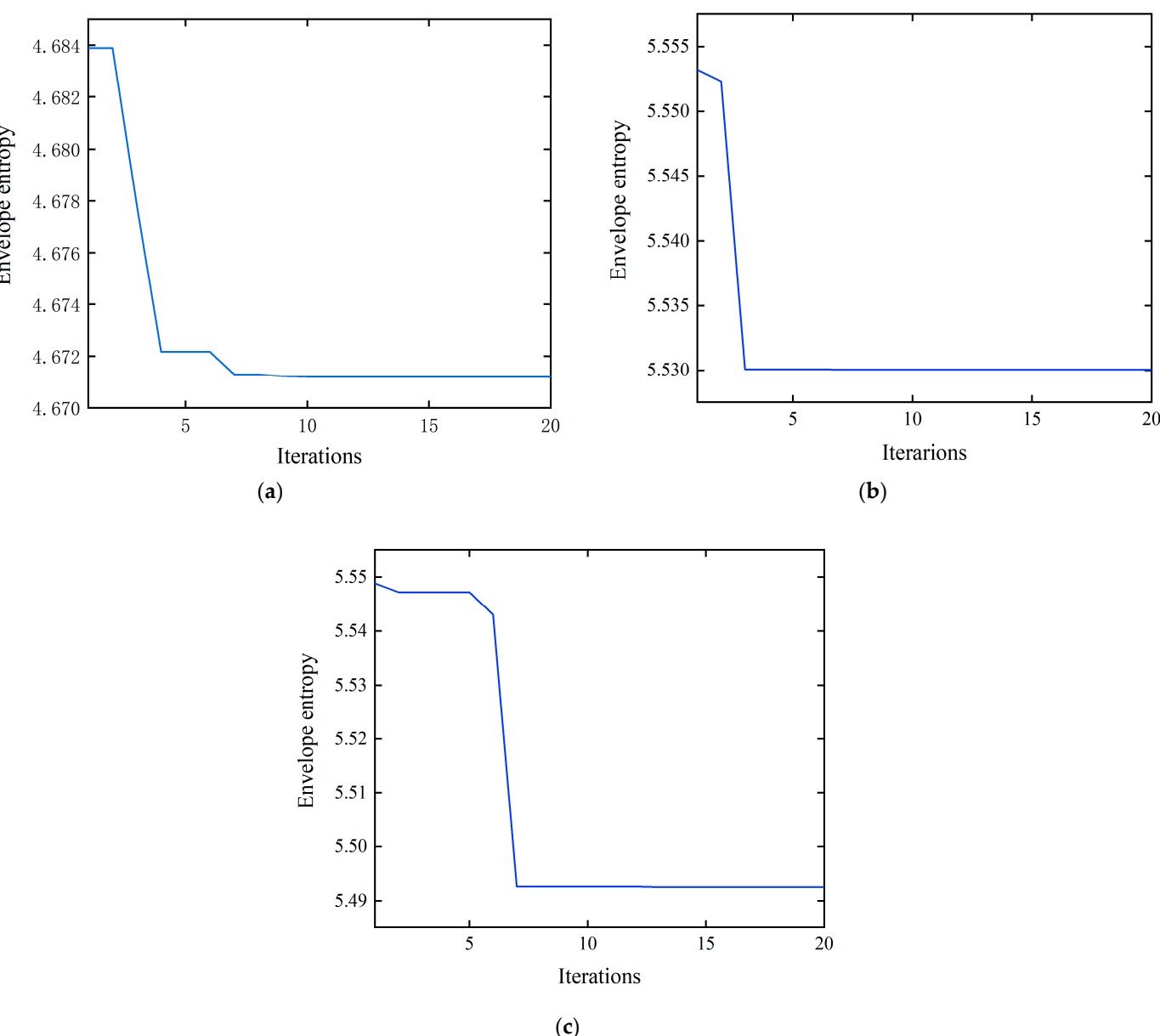

**Figure 7.** Envelope entropy iteration process. (**a**) Envelope entropy on sunny days. (**b**) Envelope entropy on cloudy days. (**c**) Envelope entropy on rainy days.

**Table 1.** The optimal parameters of IVMD.

| Weather Types | Minimum Envelope Entropy | $K$ | $\alpha$ |
|---|---|---|---|
| Sunny day | 4.6712 | 3 | 925 |
| Cloudy day | 5.5301 | 6 | 59 |
| Rainy day | 5.4925 | 7 | 93 |

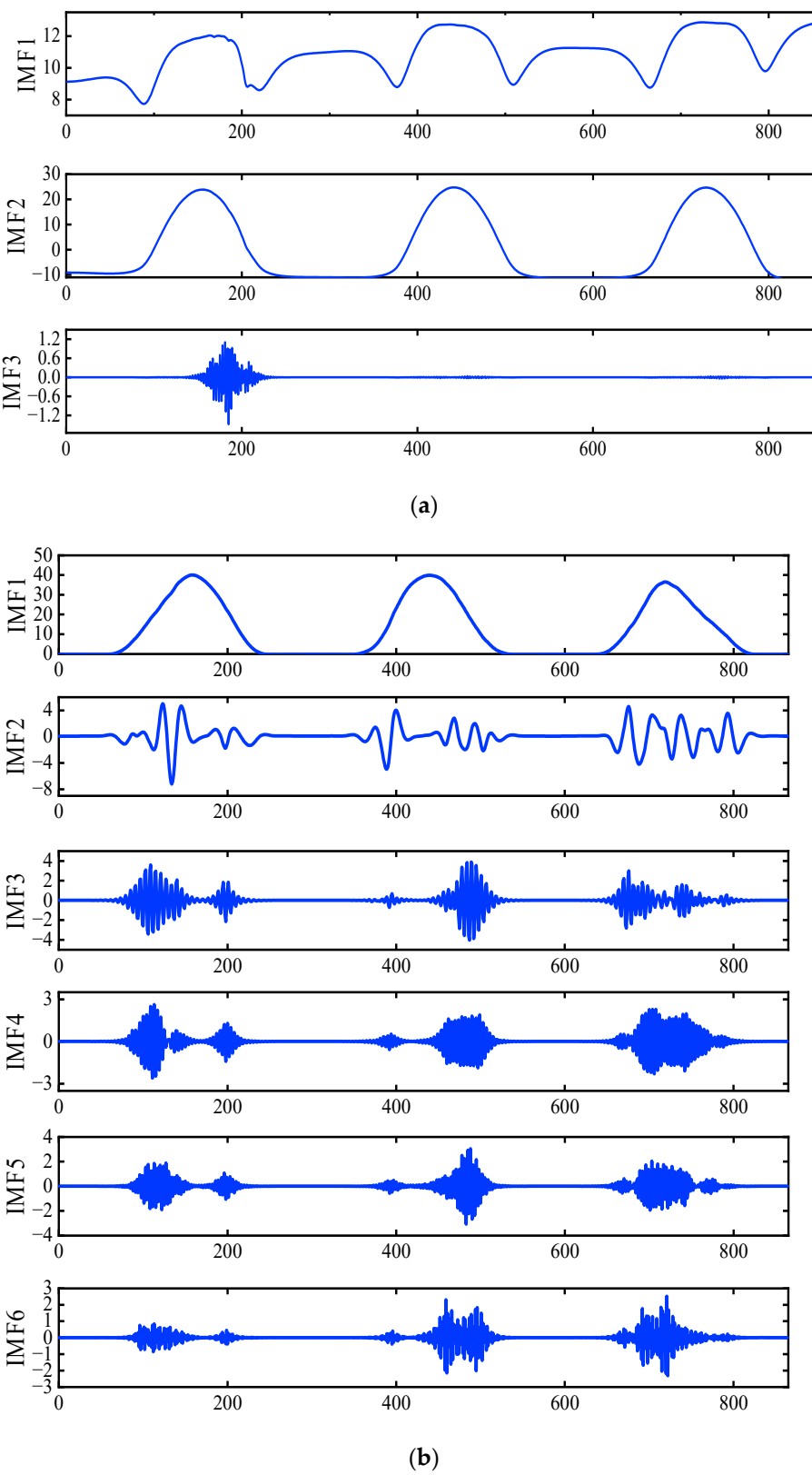

(**a**)

(**b**)

**Figure 8.** *Cont.*

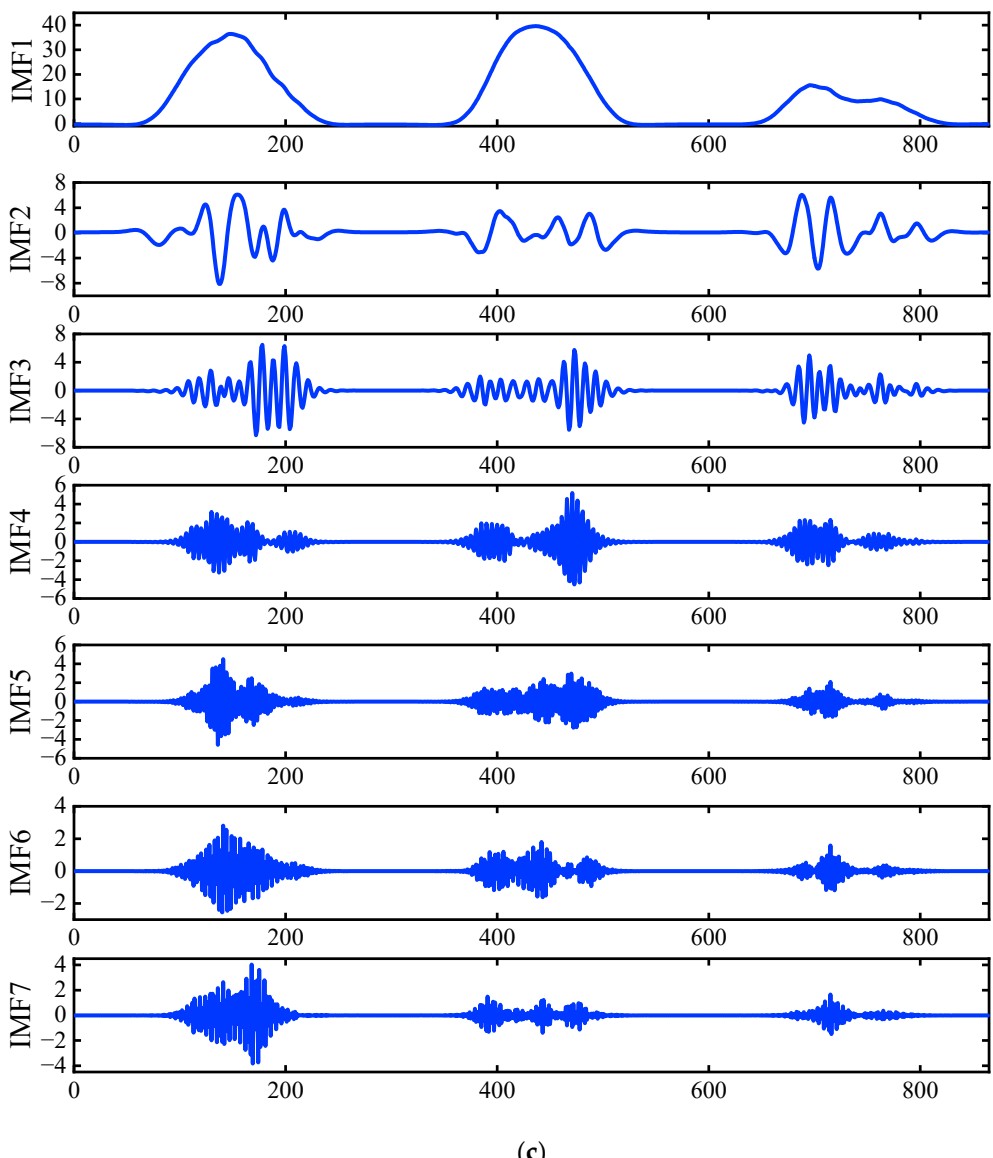

(**c**)

**Figure 8.** Photovoltaic power in IVMD decomposition. (**a**) Photovoltaic power IVMD decomposition on sunny days. (**b**) Photovoltaic power IVMD decomposition on cloudy days. (**c**) Photovoltaic power IVMD decomposition on rainy days.

Taking photovoltaic power in sunny days as an example, the comparative experimental results of TCN-GRU, IVMD-TCN-GRU, and IVMD-SSA-TCN-GRU are shown in Figure 10. In addition, in Table 2, RMSE, MAE, time consumption are evaluated to quantitatively compare the performance for each model. It can be observed that, in comparison with TCN-GRU, RMSE and MAE of the IVMD-SSA-TCN-GRU forecasting strategy are reduced by 34.1% and 36.3%, respectively. This is because non-stationary and nonlinear characteristics of photovoltaic power are weakened by photovoltaic power decomposition. Compared with the IVMD-TCN-GRU model, RMSE and MAE decrease by 16.7% and 5.2%, respectively. Because the optimal parameter combination is matched using the SSA for the TCN-GRU network, it can better preserve the original information when processing high-dimensional data, reduce the original data sequence complexity, alleviate the time delay characteristics and the fluctuation range. It can achieve higher accuracy than other single methods. In general, the IVMD-SSA-TCN-GRU forecasting model shows stronger forecasting ability and higher forecasting accuracy.

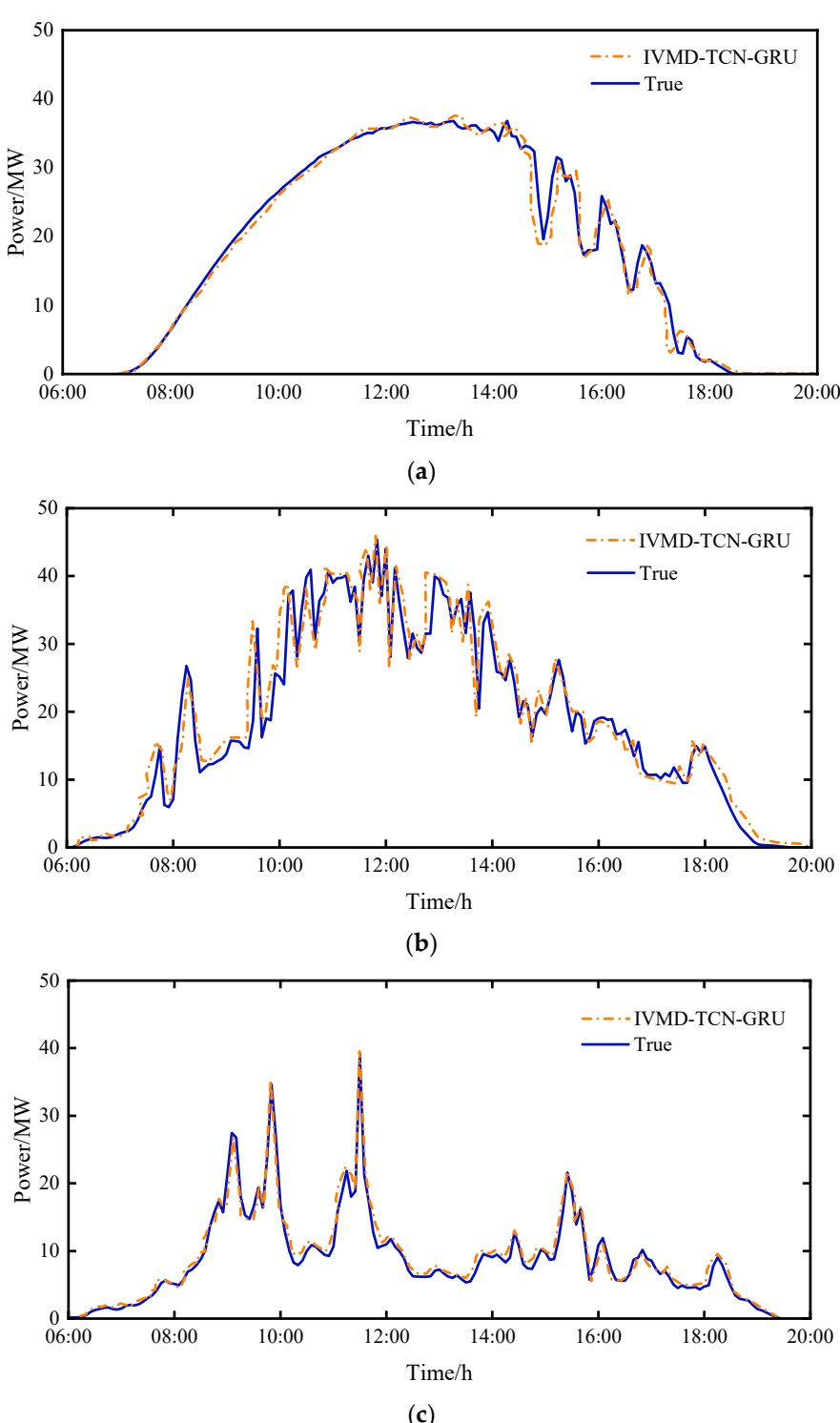

**Figure 9.** IVMD-TCN-GRU forecasting results. (**a**) Comparison diagram on sunny days. (**b**) Comparison diagram on cloudy days. (**c**) Comparison diagram on rainy days.

**Table 2.** Forecasting errors of the IVMD-SSA-TCN-GRU method.

| Evaluation Indexes | TCN-GRU | IVMD-TCN-GRU | IVMD-SSA-TCN-GRU |
|:---:|:---:|:---:|:---:|
| RMSE | 1.7479 | 1.3832 | 1.152 |
| MAE | 1.4819 | 0.9968 | 0.94461 |
| Time | 0.0141 | 0.0132 | 0.0047 |

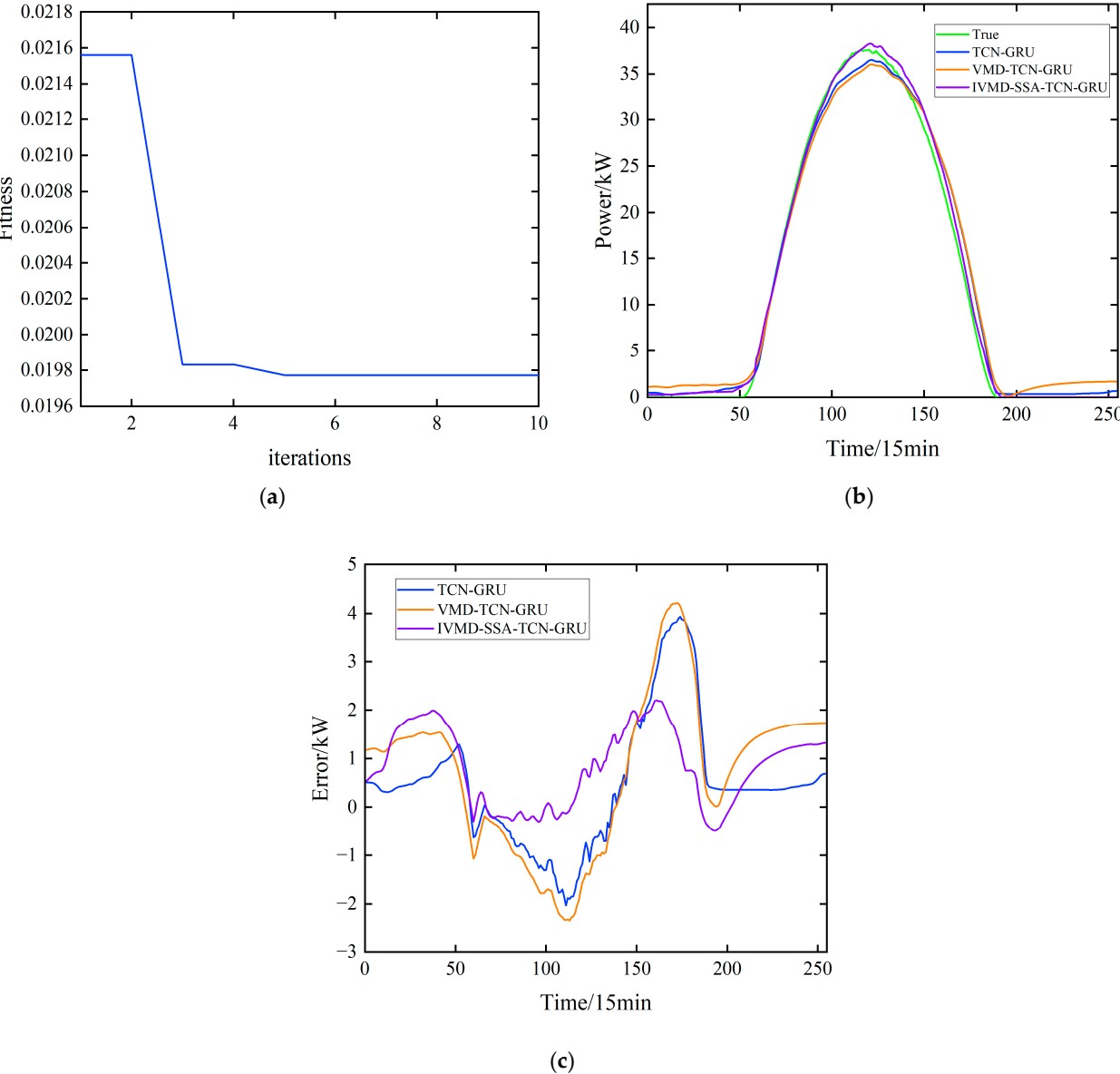

**Figure 10.** Comparison of photovoltaic power forecasting on sunny days. (**a**) The iteration of SSA-TCN-GRU on sunny days. (**b**) Photovoltaic power forecasting on sunny days. (**c**) The forecasting error on sunny days.

Moreover, the EMD-SSA-TCN-GRU and IVMD-SSA-Elman models are used as comparisons to verify the superiority under the conditions of cloudy and rainy days with large fluctuations in photovoltaic power. Figure 11 shows the forecasting results with cloudy conditions; under rainy conditions, the forecasting results are illustrated in Figure 12. The performance indexes of the different approaches are exhibited in Table 3. The proposed forecasting model has certain improvement in both RMSE and MAE, verifying the good effect of preserving environmental characteristics on photovoltaic weather type classification and model establishment. Meanwhile, VMD is applied to decompose photovoltaic power, and a forecasting model is set up for each mode before reconstruction, reducing amount of data and shortening the forecasting time. The proposed model has stronger forecasting ability and higher forecasting accuracy compared with other models. Moreover, compared with the EMD-SSA-TCN-GRU forecasting model, for the IVMD-SSA-TCN-GRU forecasting model, RMSE and MAE are reduced by 37.1% and 27.8%. The reason is that IVMD has better decomposition performance and is more suitable for decomposing and forecasting

photovoltaic power. Compared with IVMD-SSA-Elman, RMSE and MAE declined by 55.1% and 54.5%, respectively; due to issues when dealing with time series problems, the TCN-GRU network has better performance. Table 4 presents the performance metrics of the novel method proposed in this study alongside the approaches WOA-BiLSTM-Attention [50], LSTM-TCN [51], and CNN-GRU [52] in scenarios characterized by rainy conditions and substantial fluctuations in photovoltaic power generation. Our findings reveal that the proposed method outperforms the existing techniques in terms of predictive accuracy and dependability, as evidenced by the lower MAE and RMSE values obtained by our model.

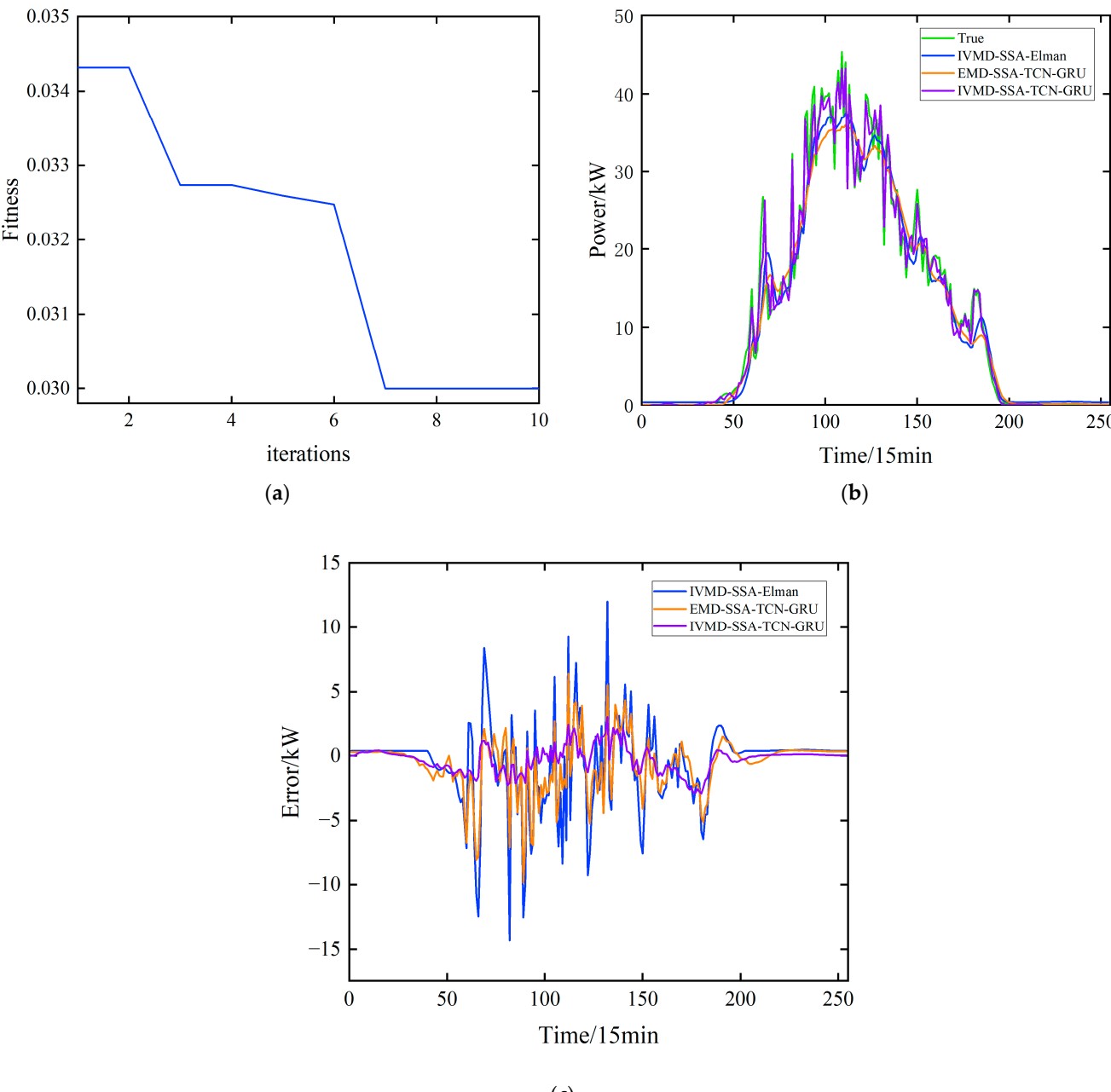

**Figure 11.** Comparison of photovoltaic power forecasting in cloudy days. (**a**) The iteration of SSA-TCN-GRU in cloudy days. (**b**) Photovoltaic power forecasting on cloudy days. (**c**) The forecasting error on cloudy days.

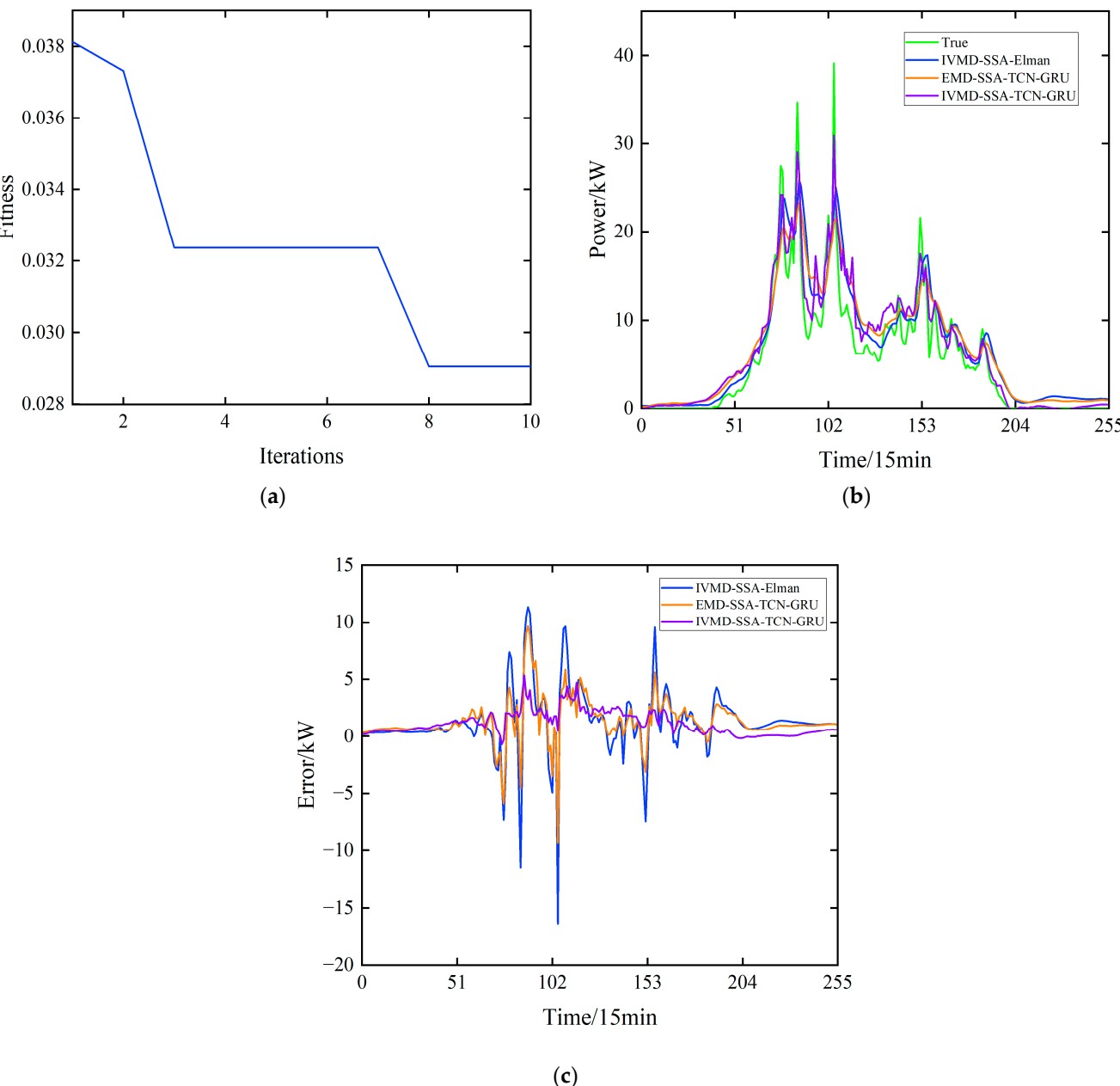

**Figure 12.** Comparison of photovoltaic power forecasting on rainy days. (**a**) The iteration of SSA-TCN-GRU on rainy days. (**b**) Photovoltaic power forecasting on rainy days. (**c**) The forecasting error in rainy days.

**Table 3.** The forecasting errors of different models.

| Weather Type | Model | MAE | RMSE |
|---|---|---|---|
| | IVMD-SSA-Elman | 2.32 | 4.17 |
| Sunny day | EMD-SSA-TCN-GRU | 2.17 | 3.76 |
| | IVMD-SSA-TCN-GRU | 1.98 | 3.41 |
| | IVMD-SSA-Elman | 3.07 | 4.84 |
| Cloudy day | EMD-SSA-TCN-GRU | 2.74 | 4.82 |
| | IVMD-SSA-TCN-GRU | 2.71 | 4.66 |
| | IVMD-SSA-Elman | 5.01 | 8.15 |
| Rainy day | EMD-SSA-TCN-GRU | 3.16 | 5.82 |
| | IVMD-SSA-TCN-GRU | 2.28 | 3.66 |

**Table 4.** The forecasting errors of different models on rainy days.

| Method | MAE | RMSE |
| --- | --- | --- |
| IVMD-SSA-TCN-GRU | 2.28 | 3.66 |
| WOA-BiLSTM-Attention | 2.45 | 3.73 |
| LSTM-TCN | 2.56 | 3.91 |
| CNN-GRU | 2.71 | 4.14 |

## 4. Discussion

The present study introduces a novel hierarchical approach to photovoltaic power forecasting that integrates IVMD into the TCN-GRU framework, further enhanced by a multi-head attention mechanism. This integration aims to tackle the inherent complexities and variabilities in PV power generation, which are significantly influenced by environmental factors. Our findings underscore the effectiveness of combining advanced signal processing techniques with deep learning models to improve the accuracy of PV power forecasts, which are crucial for the efficient management and integration of solar energy into the power grid.

The use of IVMD, optimized using the SSA, for the decomposition of PV power data marks a significant advancement in the preprocessing stage of forecasting [53–55]. This methodological choice allows for a refined extraction of the intrinsic modes within the power generation data, facilitating a more detailed and accurate analysis of the power output fluctuations. The optimization of the modal components and penalty factors through the SSA not only enhances the decomposition process but also tailors it specifically to the characteristics of the PV power data, thereby maximizing the relevance and efficiency of the subsequent forecasting model.

The core of our forecasting model combines the strengths of TCN and GRU, augmented with a multi-head attention mechanism [56–58]. This design leverages the TCN's capability to extract local feature patterns within time series data and the GRU's proficiency in capturing long-term dependencies, addressing two critical aspects of time series forecasting. The addition of a multi-head attention mechanism further elevates the model's performance by enabling a dynamic focus on the most relevant features across the time series, thereby improving the accuracy and reliability of the forecasts. This integration not only harnesses the individual strengths of these components but also mitigates their limitations, illustrating the synergistic potential of hybrid modeling approaches in complex forecasting tasks.

Incorporating environmental factors into the model represents a holistic approach to forecasting, acknowledging the significant impact of external variables on PV power output. This inclusion ensures that the model captures not only the internal dynamics of the time series data but also the influence of external conditions, providing a comprehensive framework for forecasting. The empirical validation of the model using real-world data from a PV station demonstrates its superior performance compared to traditional forecasting methods, highlighting its practical significance and potential impact on the energy sector.

However, the sophistication and computational demands of our proposed model pose challenges for real-time application and scalability. Future research could focus on optimizing the model's efficiency and exploring the feasibility of real-time forecasting, potentially broadening its applicability and utility in operational settings. Moreover, investigating the model's performance across diverse geographical locations and under varying environmental conditions would be invaluable, further affirming its robustness and adaptability. Firstly, integrating multiple components, such as the IVMD, TCN, and GRU, requires careful design and optimization to ensure the seamless functioning of the overall architecture. Coordinating the interactions between these components and fine-tuning hyperparameters can be a non-trivial task that demands computational resources and expertise. Moreover, validating the performance of the proposed model involves addressing issues related to data quality, model interpretability, and generalizability. Ensuring the robustness of the model across different datasets, geographic locations, and weather

conditions requires rigorous testing and validation procedures. Additionally, interpreting the results of the forecasting model and identifying the factors influencing its predictions can be challenging, especially when dealing with complex neural network architectures and attention mechanisms. Furthermore, the scalability and computational efficiency of the proposed model may present practical challenges, particularly when deploying the forecasting system in real-world applications with stringent latency and resource constraints. Optimizing the algorithm for efficient inference and deployment on various platforms while maintaining high forecasting accuracy is a critical consideration in operationalizing the proposed approach.

In conclusion, this study contributes a significant advancement to the field of PV power forecasting by proposing a comprehensive and integrative model that adeptly addresses the complexities of solar power generation. The innovative combination of IVMD, TCN-GRU, and a multi-head attention mechanism not only showcases the potential of hybrid models in enhancing forecast accuracy but also sets a foundation for future research aimed at optimizing and expanding the applicability of advanced forecasting techniques in the renewable energy sector.

## 5. Conclusions

In this study, we proposed a novel hierarchical forecasting model for PV power based on a multi-head attention mechanism integrated with VMD, TCN, and GRU. Through extensive experimentation and validation using real-world PV power data, we have drawn several important conclusions regarding the effectiveness and applicability of our proposed model.

(1) Our results demonstrate that the integration of VMD, TCN, GRU, and a multi-head attention mechanism significantly improves the accuracy and reliability of PV power forecasting compared to traditional methods. By leveraging VMD for signal decomposition and TCN-GRU for dynamic time series modeling, our model effectively captures both local temporal features and long-term dependencies in the data, leading to more precise predictions.

(2) The incorporation of a multi-head attention mechanism enables our model to exploit global contextual information in the time series data, further enhancing its forecasting performance. The attention mechanism allows the model to dynamically weigh the importance of different input features, thereby improving the utilization of relevant information for prediction.

(3) The optimization of VMD parameters using the SSA and the fine-tuning of GRU parameters contribute to the overall effectiveness of our proposed model. The optimization process ensures that the model is able to adapt to the specific characteristics of the input data, thereby improving its generalization capability and robustness.

Overall, our study highlights the importance of incorporating advanced machine learning techniques and considering environmental factors in PV power forecasting. The proposed hierarchical VMD-TCN-GRU multi-head attention mechanism offers a promising solution for accurately predicting PV power output, which is essential for optimizing the operation and management of solar energy systems. This research contributes to the advancement of PV power forecasting methodologies and provides valuable insights for researchers and practitioners in the field of renewable energy forecasting. The proposed model holds significant potential for facilitating the integration of solar energy into the power grid and supporting the transition towards a sustainable energy future.

**Author Contributions:** Conceptualization, H.F. and J.Z.; methodology, J.Z.; software, J.Z.; validation, J.Z. and H.F.; formal analysis, J.Z.; investigation, J.Z. and S.X.; resources, J.Z.; data curation, J.Z.; writing—original draft preparation, J.Z.; writing—review and editing, J.Z.; visualization, J.Z.; supervision, H.F.; project administration, H.F.; funding acquisition, H.F. All authors have read and agreed to the published version of the manuscript.

**Funding:** This research was funded by the National Natural Science Foundation of China, grant number 51974151.

**Data Availability Statement:** The data are unavailable due to privacy.

**Acknowledgments:** We are grateful for financial and logistical support from the H.F. Model Worker Innovation Laboratory and thank all of the original partners that supported data collection and analyses for the initial work on the Photovoltaic Power Prediction Project.

**Conflicts of Interest:** The authors declare no conflicts of interest.

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
