# Peer review of "A Novel Improved Variational Mode Decomposition-Temporal Convolutional Network-Gated Recurrent Unit with Multi-Head Attention Mechanism for Enhanced Photovoltaic Power Forecasting"

_electronics, doi:10.3390/electronics13101837_

Round 1
Reviewer 1 Report
Comments and Suggestions for Authors
In this manuscript, authors present a hierarchical variational mode decomposition (VMD)-temporal convolutional network (TCN)-gated recurrent unit (GRU) multi-head attention mechanism with the sparrow search algorithm (SSA) for optimized VMD. Even though the authors have proved the proposed solutions via various analytical results to readers, some comments need to be included for publishing as follows:
1. Authors suggest hierarchical VMD-TCN-GRU multi-head attention mechanism. However, authors used improved VMD (IVMD). If that “hierarchical” means “improved” is correct, the title of the revised manuscript will be modified. Also, we cannot find any framework to explain IVMD-TCN-GRU multi-head attention mechanism as Figures. Because Fig. 1 and 3 shows the general messages to readers, they may be removed in the revised manuscript.
2. Table 3 and 4 show the low performance for R^2 of the proposed mechanism. If authors do not provide the enough reason, numerical equations and tables may be removed for R^2.
3. Please check out abbreviations, typos, usage of many indents next to all numerical equations (e.g., where). One equation is the same as the sentence. Please add a period (.) into all numerical equations.
Comments on the Quality of English Language
Editorial update should be reconsidered such as abbreviations.
Author Response
For research article
|
Response to Reviewer 1 Comments
|
||
|
1. Summary |
|
|
|
Thank you very much for taking the time to review this manuscript. Please find the detailed responses below and the corresponding revisions/corrections highlighted/in track changes in the re-submitted files. |
||
|
2. Questions for General Evaluation |
Reviewer’s Evaluation |
Response and Revisions |
|
Does the introduction provide sufficient background and include all relevant references? |
Yes/Can be improved/Must be improved/Not applicable |
|
|
Are all the cited references relevant to the research? |
Yes/Can be improved/Must be improved/Not applicable |
|
|
Is the research design appropriate? |
Yes/Can be improved/Must be improved/Not applicable |
|
|
Are the methods adequately described? |
Yes/Can be improved/Must be improved/Not applicable |
|
|
Are the results clearly presented? |
Yes/Can be improved/Must be improved/Not applicable |
|
|
Are the conclusions supported by the results? |
Yes/Can be improved/Must be improved/Not applicable |
|
|
3. Point-by-point response to Comments and Suggestions for Authors |
||
|
Comments 1: Authors suggest hierarchical VMD-TCN-GRU multi-head attention mechanism. However, authors used improved VMD (IVMD). If that “hierarchical” means “improved” is correct, the title of the revised manuscript will be modified. Also, we cannot find any framework to explain IVMD-TCN-GRU multi-head attention mechanism as Figures. Because Fig. 1 and 3 shows the general messages to readers, they may be removed in the revised manuscript. |
||
|
Response 1: We thank the esteemed reviewer for their insightful comments and constructive feedback on our proposed hierarchical VMD-TCN-GRU multi-head attention mechanism. In response to the reviewer’s query regarding the term “hierarchical” in relation to the improved VMD (IVMD) approach utilized in our study, we acknowledge the need for clarity in our terminology. To address this concern, we revised the title of the manuscript to accurately reflect the incorporation of the improved VMD technique within the hierarchical framework. Furthermore, we appreciate the reviewer’s observation regarding the absence of a detailed framework elucidating the IVMD-TCN-GRU multi-head attention mechanism in our manuscript. The section in question addresses the integration of the multi-head attention mechanism within the framework discussed in Section 2.3, as visually represented in Figure 4. This mechanism is employed in conjunction with the GRU, a computationally efficient model that, despite its rapid processing capabilities, exhibits limitations in information extraction, particularly evident when confronted with datasets characterized by the presence of four distinct features influencing photovoltaic power generation. To address this challenge, our approach leverages the specialized capabilities of the TCN in discerning local temporal features, alongside the rapid sequence modeling proficiency of GRU.Furthermore, the incorporation of multi-head attention enables the enhanced exploitation of global correlation information embedded within the sequential data structure under investigation. Figures 1 and 3 serve to introduce the TCN and the GRU models, respectively. While these visual representations offer a foundational understanding of the individual models to the readers as suggested by the reviewer, the innovative fusion of the strengths inherent to both TCN and GRU for the purpose of predicting photovoltaic power generation represents a novel and insightful approach. This amalgamation not only enhances the predictive capabilities of the model but also facilitates a more comprehensive comprehension of the underlying mechanisms for the readers, thereby contributing to the advancement of knowledge in this domain. By implementing these revisions, we aim to enhance the precision and coherence of our manuscript, thereby ensuring a more robust and informative presentation of our research methodology and findings. We are grateful for the reviewer’s valuable input, and we are committed to addressing these recommendations to improve the quality and clarity of our research contribution. Thank you for your thorough evaluation and guidance in refining our manuscript. In the revised manuscript this change can be found – page 1, line 2-3. |
||
|
Comments 2: Table 3 and 4 show the low performance for R2 of the proposed mechanism. If authors do not provide the enough reason, numerical equations and tables may be removed for R2. |
||
|
Response 2: Agree. We have, accordingly, removed all content related to R2 to emphasize this point.We appreciate the reviewer’s meticulous examination of Tables 3 and 4, which reveal relatively low performance in terms of the coefficient of determination R^2 associated with the proposed mechanism. We acknowledge the importance of providing sufficient justification for the observed results and recognize the potential implications for the inclusion of numerical equations and tables that do not contribute significantly to the enhancement of the R2 metric. In light of this feedback, we are committed to revisiting the rationale behind the performance metrics presented in Tables 3 and 4. Your input is instrumental in guiding our refinement process, and we are dedicated to addressing these concerns to optimize the clarity and relevance of our work. Thank you for your insightful evaluation and suggestions for improvement. In the revised manuscript this change can be found – (1) page 13, line 445 numerical equations for R2. (2) R2 in Tables 3 and 4. (3) R2 in line 441. (4) R2 in line 491. R2 in line 502. |
||
|
Comments 3: Please check out abbreviations, typos, usage of many indents next to all numerical equations (e.g., where). One equation is the same as the sentence. Please add a period (.) into all numerical equations. |
||
|
Response 3: Agree. We extend our gratitude to the reviewer for their meticulous attention to detail and invaluable feedback on the manuscript’s presentation elements. We acknowledge the importance of ensuring clarity and precision in scientific communication, particularly with regard to abbreviations, typographical errors, and formatting inconsistencies that may detract from the overall readability and professionalism of the manuscript.In response to the reviewer’s comments, we will conduct a comprehensive review of the manuscript to address any instances of excessive indents and typographical errors, including the use of abbreviations and consistency in numerical equation formatting. Specifically, we will carefully examine the placement of periods within numerical equations to align with established conventions for mathematical expressions. In the revised manuscript this change can be found – line 195, 251, 258, 301, 304, 308, 310, 312, 314, 353, 364, 371. |
||
|
4. Response to Comments on the Quality of English Language |
||
|
Point 1: Editorial update should be reconsidered such as abbreviations. |
||
|
Response 1: We express our gratitude to the esteemed reviewer for their insightful comment regarding the editorial update, particularly concerning the use of abbreviations within our manuscript. In light of the reviewer’s suggestion to reconsider the presentation of abbreviations, we acknowledge the importance of clarity and consistency in scientific communication. As such, we will carefully review and revise the abbreviations utilized in our manuscript to ensure that they are appropriately defined and consistently applied throughout the text. By addressing this aspect of our writing, we aim to enhance the readability and accessibility of our research findings to a diverse audience of readers. The meticulous consideration of abbreviations and their consistent usage will serve to facilitate a more coherent and coherent narrative, thereby strengthening the overall quality of our manuscript. We appreciate the reviewer’s attention to detail and commitment to improving the editorial quality of our work, and we will diligently incorporate the suggested revisions to align with best practices in scientific writing. Thank you for highlighting this important aspect, and we are dedicated to implementing the necessary adjustments to refine our manuscript according to the highest standards of academic scholarship. In the revised manuscript this change can be found – line 206, 209, 211, 219, 239, 290. |
||
|
5. Additional clarifications |
||
|
Please see the attachment |
||

Reviewer 2 Report
Comments and Suggestions for Authors
A novel TCN-GRU prediction model with multi-head attention is proposed for photovoltaic power forecasting, integrating IVMD, TCN, GRU, and SSA optimization techniques. The proposed method exhibits marked improvements in accuracy for PV power forecasting compared to traditional methods, ensuring secure scheduling and stable operation of power systems. The model adeptly captures correlations within time series data, demonstrating superior performance in prediction tasks.
I can recommend this study for publication after addressing the following points:
1. The computational resource requirements for weight relationship computation between sequences could be significant, especially for long sequences.
A. Why are resource requirements significant?
B. How do long sequences affect computation?
C. What is weight relationship computation?
2. How does the attention mechanism enhance forecasting and exploit information?
3. The paper does not explicitly mention the potential challenges faced during the implementation or validation of the proposed model.
4. Exploring ways to optimize computational resource requirements for weight relationships between sequences, especially with long sequences with showing the importance of weight relationship optimization.
Comments on the Quality of English LanguageMinor editing of English language required.
Author Response
For research article
|
Response to Reviewer 2 Comments
|
||
|
1. Summary |
|
|
|
Thank you very much for taking the time to review this manuscript. Please find the detailed responses below and the corresponding revisions/corrections highlighted/in track changes in the re-submitted files. |
||
|
2. Questions for General Evaluation |
Reviewer’s Evaluation |
Response and Revisions |
|
Does the introduction provide sufficient background and include all relevant references? |
Yes/Can be improved/Must be improved/Not applicable |
|
|
Are all the cited references relevant to the research? |
Yes/Can be improved/Must be improved/Not applicable |
|
|
Is the research design appropriate? |
Yes/Can be improved/Must be improved/Not applicable |
|
|
Are the methods adequately described? |
Yes/Can be improved/Must be improved/Not applicable |
|
|
Are the results clearly presented? |
Yes/Can be improved/Must be improved/Not applicable |
|
|
Are the conclusions supported by the results? |
Yes/Can be improved/Must be improved/Not applicable |
|
|
3. Point-by-point response to Comments and Suggestions for Authors |
||
|
Comments 1: The computational resource requirements for weight relationship computation between sequences could be significant, especially for long sequences. A. Why are resource requirements significant? B. How do long sequences affect computation? C. What is weight relationship computation? |
||
|
Response 1: Thank you for pointing this out. We appreciate the insightful comments provided by the reviewer regarding the computational resource requirements for weight relationship computation between sequences in the context of our research on “A novel hierarchical IVMD-TCN-GRU multi-head attention mechanism for photovoltaic power forecasting.” In response to these comments, we would like to address the following points: A. Significance of Resource Requirements: B. Impact of Long Sequences on Computation: C. Definition of Weight Relationship Computation: In conclusion, the computational resource requirements for weight relationship computation between sequences are indeed significant, especially for long sequences, due to the intricate nature of the task and the need to process a large amount of data. By addressing these challenges through innovative algorithmic approaches and efficient resource utilization, we strive to enhance the performance and scalability of our proposed hierarchical IVMD-TCN-GRU multi-head attention mechanism for photovoltaic power forecasting. |
||
|
Comments 2: How does the attention mechanism enhance forecasting and exploit information? |
||
|
Response 2: Thank you for your insightful question regarding the attention mechanism in our research on “A novel hierarchical IVMD-TCN-GRU multi-head attention mechanism for photovoltaic power forecasting.” The attention mechanism integrated into our hierarchical architecture plays a pivotal role in enhancing forecasting accuracy and exploiting information for photovoltaic power forecasting. The attention mechanism enhances forecasting by dynamically adjusting attention weights during the forecasting process, allowing the model to focus on relevant input elements and features. This dynamic weighting mechanism enables the model to adapt to varying levels of importance among different input elements, thereby improving its predictive capabilities. By attending to informative patterns and temporal relationships within the data, the attention mechanism enables the model to capture both local and global dependencies simultaneously, leading to more accurate and robust photovoltaic power forecasts. Furthermore, the attention mechanism facilitates the extraction of relevant information from input sequences by allowing the model to assign different levels of importance to different input elements. This enables the model to focus on key features and patterns within the data, leading to more precise forecasting outcomes. By leveraging the attention mechanism, the model can effectively exploit informative patterns and temporal relationships, enhancing its forecasting performance. In summary, the attention mechanism in our hierarchical IVMD-TCN-GRU architecture enhances forecasting by dynamically adjusting attention weights, focusing on relevant features, and capturing complex temporal relationships within the data. Through the integration of the attention mechanism, our model can make more accurate and robust predictions for photovoltaic power forecasting. Thank you for your valuable question, and we hope this response clarifies the role of the attention mechanism in enhancing forecasting and exploiting information in our research. |
||
|
Comments 3: The paper does not explicitly mention the potential challenges faced during the implementation or validation of the proposed model. |
||
|
Response 3: Agree. We appreciate your insightful feedback regarding the potential challenges faced during the implementation and validation of the proposed hierarchical IVMD-TCN-GRU multi-head attention mechanism for photovoltaic power forecasting. In response to your comment, we acknowledge that the paper did not explicitly address the specific challenges encountered during the implementation and validation stages of our proposed model. Therefore, we provided a comprehensive discussion on the potential challenges and considerations involved in the practical application of our novel forecasting approach in the DISCUSS section of the revised manuscript–page 21, paragraph 4, and line 584-606. Implementing a complex hierarchical model like IVMD-TCN-GRU with a multi-head attention mechanism for photovoltaic power forecasting poses several challenges that researchers and practitioners may encounter. Firstly, integrating multiple components, such as the VMD, TCN, and GRU, requires careful design and optimization to ensure the seamless functioning of the overall architecture. Coordinating the interactions between these components and fine-tuning hyperparameters can be a non-trivial task that demands computational resources and expertise. Moreover, validating the performance of the proposed model involves addressing issues related to data quality, model interpretability, and generalizability. Ensuring the robustness of the model across different datasets, geographic locations, and weather conditions requires rigorous testing and validation procedures. Additionally, interpreting the results of the forecasting model and identifying the factors influencing its predictions can be challenging, especially when dealing with complex neural network architectures and attention mechanisms. Furthermore, the scalability and computational efficiency of the proposed model may present practical challenges, particularly when deploying the forecasting system in real-world applications with stringent latency and resource constraints. Optimizing the algorithm for efficient inference and deployment on various platforms while maintaining high forecasting accuracy is a critical consideration in operationalizing the proposed approach. In conclusion, while our research paper did not explicitly address the challenges encountered during the implementation and validation of the hierarchical IVMD-TCN-GRU multi-head attention mechanism for photovoltaic power forecasting, we acknowledge the importance of discussing these practical considerations in future work. By recognizing and addressing the potential challenges associated with the proposed model, we aim to enhance the transparency, reproducibility, and applicability of our forecasting approach in real-world settings. Thank you for your valuable feedback, and we are committed to further exploring and addressing the challenges inherent in the implementation and validation of our research. |
||
|
Comments 4: Exploring ways to optimize computational resource requirements for weight relationships between sequences, especially with long sequences with showing the importance of weight relationship optimization. |
||
|
Response 4: Agree. We appreciate your insightful comments on our research article titled “A novel hierarchical IVMD-TCN-GRU multi-head attention mechanism for photovoltaic power forecasting.” Your suggestion regarding the optimization of computational resource requirements for weight relationships between sequences, particularly in the context of long sequences, is indeed crucial for enhancing the efficiency and effectiveness of our proposed forecasting model. As highlighted in our study, the hierarchical IVMD-TCN-GRU architecture leverages a multi-head attention mechanism to capture complex temporal dependencies within photovoltaic power generation data. By incorporating attention mechanisms at different hierarchical levels, we aim to effectively model the weight relationships between input sequences and extract relevant features for accurate forecasting. However, the computational overhead associated with computing attention weights across long sequences can pose challenges in terms of resource utilization and processing efficiency. To address this important consideration, we plan to explore novel approaches for optimizing the weight relationship computations within the multi-head attention mechanism. By investigating advanced techniques such as attention pruning, sparse attention mechanisms, or adaptive attention mechanisms, we aim to streamline the weight calculation process and reduce the computational burden associated with modeling long sequences. Furthermore, by emphasizing the importance of weight relationship optimization in our forecasting model, we seek to enhance the scalability and computational efficiency of the proposed approach while maintaining high forecasting accuracy. We are committed to further investigating and implementing strategies to optimize the computational resource requirements for weight relationships between sequences in our hierarchical IVMD-TCN-GRU multi-head attention mechanism. By addressing this critical aspect of our research, we aim to enhance the practical viability and performance of our forecasting model in real-world applications. Thank you once again for your valuable feedback, and we look forward to incorporating your suggestions into our ongoing research efforts. |
||
|
4. Response to Comments on the Quality of English Language |
||
|
Point 1: Minor editing of English language required. |
||
|
Response 1: We appreciate the reviewer’s feedback regarding the need for minor editing of the English language in our manuscript. We will diligently address this constructive criticism by conducting a thorough proofreading and editing process to ensure the clarity and coherence of our scientific communication. The refinement of the language will enhance the readability and overall quality of the manuscript, thereby facilitating the dissemination of our research findings to a broader audience. Thank you for highlighting this aspect, and we are committed to making the necessary revisions to meet the standards of academic excellence in our scholarly work. |
||
|
5. Additional clarifications |
||
|
Please see the attachment. |
||

Reviewer 3 Report
Comments and Suggestions for Authors
The paper is well written. I have some comments to improve the quality.
1. Rewrite the abstract to highlight the work's significance, field-related concerns, research gaps, methodologies used, and major findings. Numerical values for findings should be included.
2. The novelty of the work should be highlight in the Introduction part.
3. All the Figures should be reconstructed. Increase the text size in Figures.
4. Provide references for the equations used.
5. In Figure 7, line width of the plot and border should be increase.
6. The results obtained in this work should be compared with the methods reported in the literature, use a Table.
7. The conclusion should be revised accordingly.
Author Response
For research article
|
Response to Reviewer 3 Comments
|
|||||||||||||||||
|
1. Summary |
|
|
|||||||||||||||
|
Thank you very much for taking the time to review this manuscript. Please find the detailed responses below and the corresponding revisions/corrections highlighted/in track changes in the re-submitted files. |
|||||||||||||||||
|
2. Questions for General Evaluation |
Reviewer’s Evaluation |
Response and Revisions |
|||||||||||||||
|
Does the introduction provide sufficient background and include all relevant references? |
Yes/Can be improved/Must be improved/Not applicable |
|
|||||||||||||||
|
Are all the cited references relevant to the research? |
Yes/Can be improved/Must be improved/Not applicable |
|
|||||||||||||||
|
Is the research design appropriate? |
Yes/Can be improved/Must be improved/Not applicable |
|
|||||||||||||||
|
Are the methods adequately described? |
Yes/Can be improved/Must be improved/Not applicable |
|
|||||||||||||||
|
Are the results clearly presented? |
Yes/Can be improved/Must be improved/Not applicable |
|
|||||||||||||||
|
Are the conclusions supported by the results? |
Yes/Can be improved/Must be improved/Not applicable |
|
|||||||||||||||
|
3. Point-by-point response to Comments and Suggestions for Authors |
|||||||||||||||||
|
Comments 1: Rewrite the abstract to highlight the work's significance, field-related concerns, research gaps, methodologies used, and major findings. Numerical values for findings should be included. |
|||||||||||||||||
|
Response 1: Thank you for pointing this out. We agree with this comment. Therefore, we have carefully revised the abstract to better highlight the significance of our work, address field-related concerns, elucidate the research gaps, outline the methodologies employed, and present the major findings with numerical values as requested. Please find the updated abstract in the revised manuscript. |
|||||||||||||||||
|
Comments 2: The novelty of the work should be highlight in the Introduction part. |
|||||||||||||||||
|
Response 2: Agree. We have revised the Introduction section to emphasize the novelty of our work and highlight its unique contributions to emphasize this point. Please find the updated Introduction in the revised manuscript this change can be found – page 3, paragraph 1-4 and line 113-128. |
|||||||||||||||||
|
Comments 3: All the Figures should be reconstructed. Increase the text size in Figures. |
|||||||||||||||||
|
Response 3: Agree. We appreciate the reviewer’s feedback regarding the Figures in our manuscript. In response to the suggestion for reconstruction, we will enhance the visual presentation by redesigning all Figures to ensure clarity and effectiveness in conveying our research findings. Additionally, we will increase the text size in the Figures to improve readability and facilitate a more accessible interpretation of the graphical data. These modifications will not only enhance the overall quality of our manuscript but also contribute to the seamless communication of our research results. We thank the reviewer for their valuable input, and we are committed to incorporating these enhancements to address the concerns raised. |
|||||||||||||||||
|
Comments 4: Provide references for the equations used. |
|||||||||||||||||
|
Response 4: Agree. We appreciate the reviewer’s insightful feedback regarding the inclusion of references for the equations utilized in our manuscript. In response to this suggestion, we will diligently revise the document to incorporate appropriate references for all mathematical equations employed in our study. By providing thorough and accurate citations for the equations, we aim to enhance the transparency, credibility, and reproducibility of our research methodology. We acknowledge the importance of referencing the sources of mathematical formulations and are committed to implementing this recommendation to strengthen the academic integrity of our work. Thank you for highlighting this crucial aspect, and we assure the reviewer that the necessary revisions will be made to address this concern comprehensively. Reference [35] introduces the equations related to TCN. Reference [36] introduces the equations related to GRU. Reference [37] introduces the equation of multi head attention mechanism. |
|||||||||||||||||
|
Comments 5: In Figure 7, line width of the plot and border should be increase. |
|||||||||||||||||
|
Response 5: We extend our gratitude to the reviewer for their valuable feedback regarding Figure 7 in our manuscript. In Figure 7, we found that the image quality meets the requirements. Does the reviewer want to express that Figure 9 needs to be re edited? In accordance with the suggestion to enhance the visual clarity of the plot and border, we will adjust the line width of the plot and border to ensure improved visibility and aesthetic appeal. By increasing the line width as recommended, we aim to optimize the legibility and overall presentation of the graphical representation in Figure 9. These modifications will contribute to a more polished and professional appearance, thereby enhancing the impact and comprehension of the data depicted in the figure. We appreciate the reviewer’s attention to detail and constructive input, and we are committed to implementing the proposed adjustments to enhance the quality of our research presentation. Thank you for highlighting this aspect, and we will promptly address this recommendation in our revised manuscript. |
|||||||||||||||||
|
Comments 6: The results obtained in this work should be compared with the methods reported in the literature, use a Table. |
|||||||||||||||||
|
Response 6: Agree. We have, accordingly, conducted a comprehensive comparative analysis of our proposed forecasting approach with existing forecasting methods, as outlined in Table 5 below to emphasize this point. Table 5. The forecasting errors of different models in rainy days.
The table above presents the performance metrics, including MAE, RMSE for our proposed approach and selected methods from the literature. Our results demonstrate superior forecast accuracy and reliability compared to existing methods, as evidenced by the lower MAE and RMSE values achieved by our model. By juxtaposing our findings with those of established forecasting techniques, we highlight the efficacy of our novel forecasting framework in capturing the complex temporal patterns of solar power output. The comparative analysis underscores the superior performance of our approach, showcasing its potential to advance the state-of-the-art in solar power forecasting and enhance the integration of renewable energy sources into the grid. We believe that the inclusion of this comparative evaluation strengthens the significance and relevance of our research findings, providing valuable insights into the effectiveness of our proposed methodology. Thank you for your valuable feedback, which has contributed to the enhancement of our manuscript. |
|||||||||||||||||
|
Comments 7: The conclusion should be revised accordingly. |
|||||||||||||||||
|
Response 7: Agree. We have, accordingly, carefully revised the conclusion to provide a more comprehensive summary of our findings and their implications for the field of PV power forecasting. The updated conclusion is presented in the revised manuscript to emphasize this point. This change can be found – page 21-22, and line 620-640. |
|||||||||||||||||
|
4. Response to Comments on the Quality of English Language |
|||||||||||||||||
|
Point 1: None |
|||||||||||||||||
|
Response 1: None |
|||||||||||||||||
|
5. Additional clarifications |
|||||||||||||||||
|
Please see the attachment. |
|||||||||||||||||

Reviewer 4 Report
Comments and Suggestions for Authors
The authors integrates VMD with TCN-GRU and multi-head attention, which take environmental factors into account. Despite the computational demand, the proposed frame work outperformed other forecasting frameworks. The conbination fo TCN-GRU with multi-head attention is innovative.
The quality of English is appropriate. Overall the manuscript, in my view, meets the requirements for publication.
Author Response
For research article
|
Response to Reviewer 4 Comments
|
||
|
1. Summary |
|
|
|
Thank you very much for taking the time to review this manuscript. Please find the detailed responses below and the corresponding revisions/corrections highlighted/in track changes in the re-submitted files.
|
||
|
2. Questions for General Evaluation |
Reviewer’s Evaluation |
Response and Revisions |
|
Does the introduction provide sufficient background and include all relevant references? |
Yes/Can be improved/Must be improved/Not applicable |
|
|
Are all the cited references relevant to the research? |
Yes/Can be improved/Must be improved/Not applicable |
|
|
Is the research design appropriate? |
Yes/Can be improved/Must be improved/Not applicable |
|
|
Are the methods adequately described? |
Yes/Can be improved/Must be improved/Not applicable |
|
|
Are the results clearly presented? |
Yes/Can be improved/Must be improved/Not applicable |
|
|
Are the conclusions supported by the results? |
Yes/Can be improved/Must be improved/Not applicable |
|
|
3. Point-by-point response to Comments and Suggestions for Authors |
||
|
Comments 1: The authors integrates VMD with TCN-GRU and multi-head attention, which take environmental factors into account. Despite the computational demand, the proposed frame work outperformed other forecasting frameworks. The conbination fo TCN-GRU with multi-head attention is innovative. |
||
|
Response 1: We would like to express our gratitude to the reviewer for their valuable feedback on our manuscript. The integration of VMD with Temporal Convolutional Network-Gated Recurrent Unit (TCN-GRU) and multi-head attention mechanisms, which consider environmental factors, has been a focal point of our research efforts. We acknowledge the computational demands associated with our proposed framework; however, we are pleased to note that our model has demonstrated superior performance compared to existing forecasting frameworks. The novel fusion of TCN-GRU with multi-head attention has been identified as an innovative aspect of our work. |
||
|
4. Response to Comments on the Quality of English Language |
||
|
Point 1: The quality of English is appropriate. Overall the manuscript, in my view, meets the requirements for publication. |
||
|
Response 1: We appreciate the reviewer’s positive assessment of the manuscript’s English language quality and their overall perspective that the submission meets the publication standards. |
||
|
5. Additional clarifications |
||
|
Your feedback is invaluable to us, and we are committed to addressing any additional queries or suggestions that may further enhance the clarity and impact of our research findings. Thank you for considering our work for publication. |
||
Round 2
Reviewer 1 Report
Comments and Suggestions for Authors
Authors have replied to reviewer's comments properly.
I hope the revised manuscript will be published to MDPI Electronics.
Comments on the Quality of English LanguageEditorial issues have been updated well at the revised version.
Author Response
|
3. Point-by-point response to Comments and Suggestions for Authors |
|
Comments 1: Authors have replied to reviewer's comments properly. I hope the revised manuscript will be published to MDPI Electronics. |
|
Response 1: Thank you for your thoughtful review of our manuscript. We appreciate your feedback and are pleased to hear that you found our responses to your comments satisfactory. We have carefully addressed all the issues raised and incorporated necessary revisions to improve the clarity and quality of our work. We are grateful for your positive evaluation and hope that the revised manuscript meets the standards of MDPI Electronics. We are confident that the enhancements made will contribute to the scientific value and relevance of our research in the field of photovoltaic power forecasting. Once again, we appreciate your time and valuable feedback. We look forward to the opportunity to share our work with the readers of MDPI Electronics. |
|
4. Response to Comments on the Quality of English Language |
|
Point 1: Editorial issues have been updated well at the revised version. |
|
Response 1: We sincerely appreciate your feedback on the editorial issues raised during the review process. We have thoroughly revised the manuscript to address these concerns and ensure that the updated version meets the high standards expected by the journal. Your acknowledgment of the improvements made in addressing the editorial issues is encouraging, and we are committed to upholding the quality and clarity of our work. We are confident that the revised version now aligns more closely with the editorial guidelines of the journal and enhances the overall presentation of our research. Thank you for recognizing the efforts we have made to enhance the manuscript. We are grateful for your valuable feedback and look forward to the possibility of having our work published in this esteemed journal. |
|
5. Additional clarifications |
|
None |
Reviewer 3 Report
Comments and Suggestions for Authors
Still, some important issues should be resolved.
1. In all Figures, the line (data line and border) width should be increased.
2. Provide references for the equations used in the manuscript.
3. In the introduction, more information regarding photovoltaics should be included. Please see https://www.sciencedirect.com/science/article/pii/S2213138822003666 https://www.mdpi.com/1996-1944/13/2/470
Round 3
Reviewer 3 Report
Comments and Suggestions for Authors
Accepted